**Investigation**

# Dimensionality reduction of genetic data using contrastive learning

Filip Thor 🆔,* Carl Nettelblad 🆔

Division of Scientific Computing, Department of Information Technology, Science for Life Laboratory, Uppsala University, Uppsala SE-752 37, Sweden

*Corresponding author: Division of Scientific Computing, Department of Information Technology, Science for Life Laboratory, Uppsala University, Uppsala SE-752 37, Sweden. Email: filip.thor@it.uu.se

We introduce a framework for using contrastive learning for dimensionality reduction on genetic datasets to create principal component analysis (PCA)-like population visualizations. Contrastive learning is a self-supervised deep learning method that uses similarities between samples to train the neural network to discriminate between samples. Many of the advances in these types of models have been made for computer vision, but some common methodology does not translate well from image to genetic data. We define a loss function that outperforms loss functions commonly used in contrastive learning, and a data augmentation scheme tailored specifically towards SNP genotype datasets. We compare the performance of our method to PCA and contemporary nonlinear methods with respect to how well they preserve local and global structure, and how well they generalize to new data. Our method displays good preservation of global structure and has improved generalization properties over t-distributed stochastic neighbor embedding, Uniform Manifold Approximation and Projection, and popvae, while preserving relative distances between individuals to a high extent. A strength of the deep learning framework is the possibility of projecting new samples and fine-tuning to new datasets using a pretrained model without access to the original training data, and the ability to incorporate more domain-specific information in the model. We show examples of population classification on two datasets of dog and human genotypes.

Keywords: deep learning; machine learning; PCA; dimensionality reduction; population genetics

## Introduction

With the advent of next-generation sequencing techniques, the number of genomes sequenced annually is increasing exponentially as the cost continues to drop (Stephens *et al.* 2015). The continued expansion in available data has made analyzing genetic datasets using deep learning (DL) approaches more attractive than before due to their dependence on the availability of large and rich datasets. Neural networks have been employed with great success in many fields but have only seen limited use and success in genomics.

In this work, we are concerned with dimensionality reduction of genotype data. Dimensionality reduction methods have been broadly used for analyzing population structure by giving a way to visualize similarities and relatedness of populations. Existing tools have helped bridge gaps in historical knowledge and data on, e.g. human migration and admixture events (Lazaridis *et al.* 2014). It has also been used in breeding programs, linking current lines to historical lines and landraces (Pincot *et al.* 2021). We consider single nucleotide polymorphism (SNP) genotype data, and transforming the high-dimensional SNP vectors down to two dimensions yields an understandable visualization of the individuals in a dataset. Classically, this has been done using principal component analysis (PCA), which is a method that transforms the data using orthogonal linear combinations of the inputs such that the variance of the data is maximized in the first dimensions (Pearson 1901). PCA transformations of genetic variant data

have been shown to correlate to the geographic origin of samples from natural populations (Novembre *et al.* 2008). PCA is a linear model, and a trained PCA embedding can be used to embed new samples with reference to the training population. This has, for example, been used to project ancient samples onto an embedding trained on modern individuals to investigate the origin of ancient samples (Rodríguez-Varela *et al.* 2023).

Other contemporary methods of dimensionality reduction have become more popular for visualization of population structure. Examples of this are t-distributed stochastic neighbor embedding (t-SNE) (van der Maaten and Hinton 2008) and Uniform Manifold Approximation and Projection (UMAP) (McInnes *et al.* 2018). They are both neighbor-based graph learning algorithms that attempt to find a low-dimensional embedding that has topological structures similar to the high-dimensional input data and specifically try to preserve local relationships in the data.

For as long as these nonlinear methods have been in use, their use has also been contested. While they can group samples with similar features; they, unlike PCA, do not preserve an interpretable distance between samples in the embedding. The publication of a flagship paper summarizing the genomic data in the All of Us Research Program (Bick *et al.* 2024) became a nexus for this (Kozlov 2024; Marx 2024), due to the use of UMAP with settings that promoted well-contained clusters, and then overlaying survey-reported race and ethnicity label data in that embedding in a way that made some of these seem well-separated. One vocal

voice in the initial critique was Jonathan Pritchard of Stanford University, who in a thread on X wrote, among other things: "[…]The UMAP algorithm, by design, exaggerates the distinctiveness of the most frequent ancestries, a message that can be misinterpreted by the public. UMAP pulls unusual genotypes towards the majority clusters; in particular it fails to represent admixture in a sensible way (admixture is fundamentally additive, while UMAP is not). In this setting, the messiness of Admixture or PCA plots yield a better reflection of the data[…]" (Pritchard 2024).

Both UMAP and t-SNE focus on preserving the local vicinity of an individual. In a human genetic context, dense clusters can sometimes be formed by chains of individuals being remote cousins, while not necessarily respecting ancestral population structure. More importantly, as noticed in the critique by Pritchard and others, the methods can exaggerate some aspects of distinctiveness—creating a situation where an admixed individual ends up as part of the cluster it is most similar to, while simultaneously imposing a false separation to other clusters with significant affinity. This may distort the data and fail to capture admixture events in a representative way.

We want to emphasize that these are different ways of visualizing the data, and the notion of one method being the best is determined by which characteristics of the data we want the embedding to retain. Nonetheless, we believe it is reasonable to require that genetically highly similar individuals should always end up close to each other in an embedding intended for general population structure. An embedding that preserves some cliques of connectedness while putting some individuals with a high degree of identity by state at a great distance to each other, would tend to be problematic in many cases. We argue that UMAP and t-SNE will tend to produce such embeddings, when used with settings that do well on other quality metrics.

Regarding labels, we believe that it is useful to use demographic data when trying to understand population structure, and to try to assess the possible under-representation of certain communities in health-related resources such as All of Us. However, the combination of a coarse categorization and an embedding promoting disconnected clusters, was especially fraught. Using a finer categorization can sometimes help illustrate how arbitrary the exact boundaries between groupings can be. While methods that strictly preserve additivity, such as PCA, can be useful in reducing the issues of distorting the connectedness in the original data, they can also sometimes be limiting.

In this context, another approach to dimensionality reduction, and the one we take here, is using neural networks. Using neural networks in a genomic setting is not unique and is getting more traction lately, including in applications to analyze population structure.

Most of the existing work on using neural networks to visualize population structure utilizes what are called autoencoders (Battey et al. 2021; Ausmees and Nettelblad 2022). These models are trained to encode the input SNPs into a lower-dimensional representation, and from this encoding reconstruct the input. We build on the work by Ausmees and Nettelblad (2022) and use convolution layers in the neural networks which in theory enable our models to capture linkage between base pair coordinates.

One benefit of neural network models over neighbor-based ones is in regards to using a trained model on unseen data. The Python implementations OpenTSNE and umap-learn both have the ability to first train an embedding, and then project new samples. However, both implementations save the raw input data, since projecting the new samples consists of comparing the unseen samples with the training data, both in genotype space and in embedding space. This means that to project new samples, access to the training data will be needed. This is a drawback if the training data is large, or sensitive. An advantage of neural networks is that one can first train on a larger dataset. Later fine-tuning can be done starting out from pretrained weights, rather than the full data. While neural network weights are not guaranteed to preserve the privacy of individuals used in training the model, it is still preferable compared to distributing the original data from both a scalability and privacy standpoint.

In this work, we propose the use of contrastive learning. This method also encodes the data into a lower-dimensional space but is not trained to reconstruct the input. The model is instead trained using a loss function applied directly to the encoding space, where the loss function considers relations between different samples in their embedding coordinates. Key components here are specialized loss functions, and data augmentation (Chen et al. 2020). Augmentation has proven important in many applications of deep learning, including computer vision, and we apply similar techniques in an appropriate way to genomic data. One previous attempt to use contrastive learning on genomic data used a shallow network with two hidden layers and an augmentation scheme that consisted of changing some homozygous variants to heterozygous (Ubbens et al. 2022). Inspired by this, we adopt a version of their marker-flipping strategy in this study. In addition, we use a more complex neural network architecture designed to capture linkage, an augmentation scheme based on missing data, and a new loss function.

When analyzing population structure, it is desirable that an embedding will reflect both global and local structure well. If we can retain much of the vital information in two dimensions, we could assume that a higher-dimensional embedding dimension would retain more information. This is the goal for the types of models we develop—we want a foundational model trained to understand the genomic structure of the organism under consideration. With a trained model that can produce a high dimensional embedding, one can employ techniques like fine-tuning and transfer learning to tailor the model to the application at hand, whether that be, e.g. population structure analysis, identification of causal variants, or genomic prediction.

Linkage information is not captured in methods like PCA or t-SNE since they consider each variant in isolation. A neural network model that incorporates convolutional layers can, in theory, capture linkage patterns and epistatic effects between variants, but this effect is not studied in this work.

## Materials and methods

This section introduces the deep learning setting and the main components of our contrastive learning framework. We also describe the datasets analyzed and the performance metrics used.

### Deep learning

In a general machine learning setting, we consider some dataset $\mathcal{D} = \{\mathcal{X}_i, \mathcal{Y}_i\}_{i=1}^N$ that we want to study, which consists of some samples $\mathcal{X}$, and labels of interest $\mathcal{Y}$. We then want to find the underlying relation that dictates the mapping $f : \mathcal{X} \to \mathcal{Y}$. The approach taken to understand this relation is dependent on the problem setup. We might use *supervised learning*, a case where we assume knowledge of labels $\mathcal{Y}$ on some training set, and want to find a generalizing function $\hat{f}$ that approximates $f$. An example of this is genomic prediction, where given genotypes $\mathcal{X}$ we want to predict phenotypes $\mathcal{Y}$. Or, we might want to use *unsupervised learning*, where we assume no knowledge of the training labels. An example

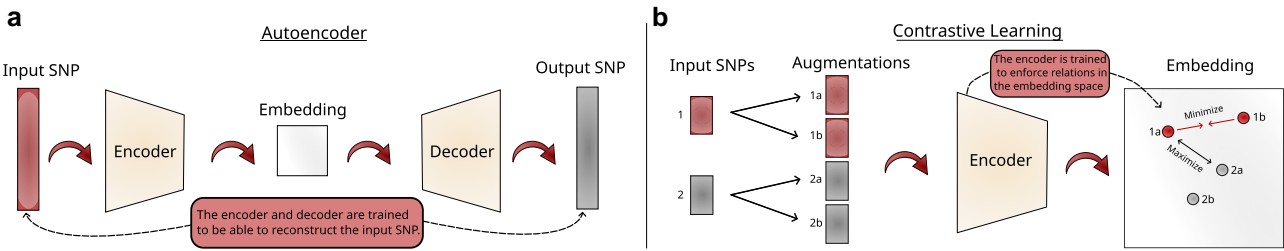

**Fig. 1.** a) Concept of an autoencoder architecture. The neural network model is trained to reconstruct the input. The latent space, or embedding, acts as a bottleneck and is used to visualize the population. b) Concept of architecture used for contrastive learning. It utilizes an encoder similar to the autoencoder but does not attempt to reconstruct the data. The model is trained by comparing the embedding coordinates of different samples. In both models, the embedding coordinates are used as the population visualization.

here would be dimensionality reduction, akin to using PCA to study population structure. Here, we only use the genotypes $\mathcal{X}$ and hope that the model can reveal insights into the population structure. When labels $\mathcal{Y}$ are available, the level of colocation of samples with identical labels in the resulting embedding can serve as a proxy for embedding quality.

Deep neural networks are a family of models with a compositional structure, where information is fed through and transformed in what are called *layers*. They sequentially process the data by taking linear combinations of the output of the previous layer and explicitly introduce nonlinearities through activation functions. This yields a nonlinear model with an explicit input–output mapping.

There have been several attempts at creating neural network models for visualizing population structure, two examples are pop-vae (Battey *et al.* 2021) and GCAE (Ausmees and Nettelblad 2022). Both are *autoencoder* architectures. This means that the neural network consists of an encoder network, which is a function $f_E : \mathbb{R}^D \to \mathbb{R}^d$ that takes the full $D$-dimensional input SNP vector, and maps it to a $d$-dimensional space, called the embedding, or latent space, which is used for the population visualization. The embedding is then transformed by a decoder network $f_D : \mathbb{R}^d \to \mathbb{R}^D$, mapping the embedding coordinates back into the $D$-dimensional space. The autoencoder network can be seen as the composition of the two networks $f_{AE} = f_D \circ f_E$ and is trained to reconstruct the input. That is, we use a loss function that captures the discrepancy between the input $x$ and the output $\hat{x} = f_{AE}(x) = f_D \circ f_E(x)$ produced by the autoencoder. One example of a loss function used in this case is the mean squared error between the true genotypes and the reconstruction. The main motivation for why the embedding dimension of an autoencoder would be a good representation of the population structure is that to reconstruct the input accurately, the lower-dimensional embedding acts as a bottle-neck and is required to extract the most distinctive, or representative, information in the data (Goodfellow *et al.* 2016). This is therefore an indirect way of letting the model know what relations to learn. An illustration of the autoencoder concept is shown in Fig. 1a.

With contrastive learning, the architecture is altered slightly. Instead of training the model to reconstruct the input, the model consists only of the encoder network $f_E$ and uses a different loss function that is applied directly to the embedding coordinates $z = f_E(x)$. Figure 1b shows an illustration of the concept behind the contrastive learning framework.

Both the autoencoder and contrastive learning are part of a class of models that are referred to as *self-supervised learning*. They define their own relations or rules from the data that the model is trained to adhere to, without any access to labels. In our work, we only use preexisting labels to evaluate the models after training.

## Contrastive learning

This section introduces the details of the contrastive learning implementation that we propose.

### Anchor, negatives, positives

In conventional supervised learning, even if samples are considered together in batches, there is no explicit within-batch interaction between them. In contrastive learning, samples and their embeddings are explicitly compared, and the network weights are updated based on the relationships of the embeddings of the samples considered.

To make the embedding comparison in each loss contribution computation, the input genotypes take on one of three different roles: *anchor*, *positive*, and *negative*. These will be denoted as $z$, $z^+$, and $z^-$, respectively.

When computing the loss function of a batch, each sample will contribute a term to the total loss where it acts as the anchor, which is the basis for each comparison. The positive is a sample we want the model to recognize as related to the anchor, and the negative is a sample to be recognized as dissimilar to the anchor. In general, we want the loss functions to train the network to map the anchor closer to the positive than to the negative.

In some loss functions, we have several negatives per anchor denoted by the set $\{z_i^-\}_{i=1}^{N-1}$. For brevity in the coming equations, we denote this set of embeddings as $Z = \{z, z^+, \{z_i^-\}_{i=1}^{N-1}\}$.

### Loss function

One basic contrastive loss function is the triplet loss function (Schroff *et al.* 2015), which uses only one negative, and has the form

$$\mathcal{L}_{\text{triplet}}(z, z^+, z^-) = \max(|z - z^+| - |z - z^-| + \alpha, 0). \tag{1}$$

That is, if the distance from the anchor to the positive is larger than the distance from the anchor to the negative (with a margin of $\alpha$), we get a nonzero loss contribution, and the network weights will be updated using the gradient of the loss function in a way that attracts the positive and repels the negative from the anchor. Figure 2a shows the distances that are considered when computing the triplet loss. If we get a zero loss, the triplet $\{z, z^+, z^-\}$ are deemed to be in a configuration that is good enough to not be corrected further.

The triplet loss illustrates the concept well but has been improved and generalized by comparing the anchor to multiple negative samples. This is referred to as the N-pair loss (Sohn 2016) and is defined in Equation (2) as

$$\mathcal{L}_{\text{N-pair}}(Z) = \log\left(1 + \sum_{i=1}^{N-1} \exp\left(z^\top z_i^- - z^\top z^+\right)\right). \tag{2}$$

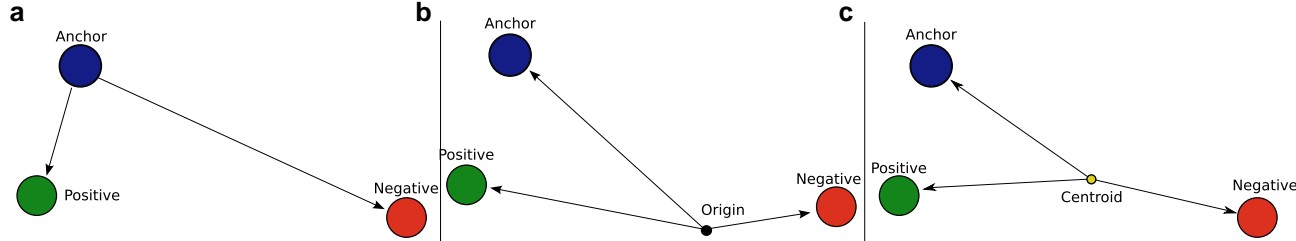

**Fig. 2.** Illustration of how samples are compared in the different contrastive loss functions. a) Triplet loss, b) N-pair loss, c) centroid-based N-pair loss. A new centroid will be computed and used for each triplet within each batch of samples.

Here, we have an expression resembling a softmax loss, which gives a larger loss contribution if the distance from the anchor to a negative is small, and weights the contribution more. Another detail here is that instead of using a distance metric like Euclidean distance, the similarity of the samples is measured by their inner product. This inner product is taken in the coordinate system defined by the global origin (0, 0), which is illustrated in Fig. 2b.

We saw that using the basic triplet loss as described in Equation (1) would yield embeddings that failed to discern population structure, with limited discriminative power and a blurry visual impression. Also, the more intricate losses, similar to the N-pair loss in Equation (2), seemed to benefit a lot in terms of training stability from using some specific model alterations, for example, forcing the output to have unit length by applying $L_2$ normalization. This was not feasible in a case where we wanted to obtain 2D visualizations. This led us to another, more localized approach we denote the centroid-based N-pair loss function, and the use of map projections, described in a later section.

### Centroid-based N-pair loss

We propose a centroid-based version of the triplet and N-pair losses, where the contribution of one anchor sample to the total loss is computed in a new reference frame for each anchor-negative pair.

Instead of computing the inner products with respect to the origin, it is performed in a new reference frame which consists of simple translation. For each negative sample, a new centroid $C_i$ is computed, which the vectors will be defined in reference to when computing the inner products. The new vector will simply be shifted by the centroid, e.g. $z_{C_i} = z - C_i$. Since we would assume that in general the positive sample will be embedded closer to the negative, we define the centroid as

$$C_i = \frac{z + 2z_i^- + z^+}{4}. \tag{3}$$

In our scheme, the positive and the anchor are two representations of the same individual. Thus, it makes sense to double the weight of the negative to find the centroid for the individuals. Supplementary Figure S4 gives an illustration of the centroid placement with and without this weighting. If the positive is not close to the anchor, this scaling means that the loss focuses more on repulsion from the negative, rather than attraction to the positive. When distances grow large, the loss function will decrease rather quickly. To control the magnitude, we also scale the vectors by the magnitude of the largest vector, computed as

$$\mu_i = \max\left(||z_{C_i}||_2^2, ||z_{C_i}^+||_2^2, ||z_{C_i}^-||_2^2\right). \tag{4}$$

Figure 2 shows illustrations that highlight the differences between the three approaches to computing the similarity used in triplet, N-pair, and our centroid version, respectively.

The centroid-based loss function is based on the N-pair loss formulation in Equation (2), with minor changes. If we introduce the following notation, we can make the expression more succinct. Let

$$\tilde{z}_i = \frac{z - C_i}{\sqrt{\mu_i}}, \tag{5}$$

then the centroid-based N-pair loss can be expressed as

$$\mathcal{L}_{\text{centroid}}(Z) = \log\left(1 + \sum_{i=1}^{N-1}\left(\frac{\exp(\tilde{z}_i^\mathsf{T}\tilde{z}_i^-)}{\exp(\tilde{z}_i^\mathsf{T}\tilde{z}_i^+)} - e^{-2}\right)\right). \tag{6}$$

The subtraction term $e^{-2}$ comes from the way the centroids are defined. The "best case" configuration of the samples would be that the anchor and positive sample would get mapped close compared to the negative. We want this to give a loss contribution of zero. The definition of the centroid in (6) and the scaling in (5) gives that $\tilde{z}_i^\mathsf{T}\tilde{z}_i^- = -1$ and $\tilde{z}_i^\mathsf{T}\tilde{z}_i^+ = 1$, and thus $\frac{\exp(\tilde{z}_i^\mathsf{T}\tilde{z}_i^-)}{\exp(\tilde{z}_i^\mathsf{T}\tilde{z}_i^+)} = \exp(-2)$ in that case. This constant offset from zero is subtracted to keep the loss unbiased with respect to the centroid definition.

The total loss for a batch consists of summing the loss contributions computed as in Equation (6) using each sample in the batch as the anchor once. The coming sections will describe how the positive and negative samples are chosen.

### Augmentations as positives

In computing the loss function, each sample in a batch will act as the anchor once. For each of these anchors, we need to assign positive and negative samples. We want a positive sample to have a different input signature to the anchor while having the same label. That is, we want the positive to be distinct from the anchor but be recognized by the model to be related. An ideal positive sample would be another individual in the same population as the anchor. They will be distinct individuals, but share similar genetic profiles. In a supervised setting, this sample could be another sample in the batch that shares the same or has a similar label as the anchor, but even when labels are available, providing them to the model might introduce unwanted bias. A common choice is to generate a new sample to use as the positive through data augmentation (Chen *et al.* 2020). For example, one study used a random flipping of homozygous sites to heterozygous (Ubbens *et al.* 2022).

There is untapped potential regarding the choice of augmentation scheme since it allows us to incorporate domain-specific knowledge. We want to alter the data in a way that should not

affect the label information in a meaningful way. The literature is rich on the topic of data augmentation in computer vision for image classification (Cubuk *et al.* 2019; Shorten and Khoshgoftaar 2019). It has been indicated that contrastive learning benefits specifically from strong data augmentation (Chen *et al.* 2020). However, augmentation strategies used in image analysis cannot be applied directly to genotype data. There are certain properties that can be exploited when it comes to images that do not translate to vectors of genotypes at predetermined SNPs. Images are generally rotational and translationally invariant—a slightly shifted and rotated image should still retain its label information. This does not hold for SNP data. Carrying an alternative allele in one SNP is a completely different thing from carrying an alternative allele in the next SNP within the set of positions used. Another common augmentation for images is mirroring it along an axis, which would be unsuitable for the same reason.

In genotyping, some of the samples will have variants with low confidence and may be marked as missing. One way to deal with this is to impute the data to try to get a more complete dataset. Due to the versatility of the deep learning approach, we can utilize the concept of missingness and incorporate it into the model. We build upon work where the input to the network is the full SNP vector with missing values set to −1, and add a mask channel that has the same dimensions as the SNP vector and is 0 everywhere, except at the missing SNP positions where it is set to 1 (Ausmees and Nettelblad 2022). This was introduced to act as a regularization technique, setting a fraction of the data as missing during training, to increase generalization performance. Here, we use missingness as a form of data augmentation when generating the positive samples. Given a SNP vector with no missing data, masking parts of the vector should not greatly affect how it is embedded. The quality of the data will be worse with lower overall information coverage, but behind the masking, it is still the same individual. We want a model that is robust to missing data, and masked versions of the same sample should be recognized as being similar. Depending on the masking rate $p_{mask}$, different samples with similar genetic makeup will be more likely to have similar augmented versions than samples of other genetic origin.

We also adopt the approach taken in Ubbens *et al.* (2022), where some variants are randomly changed. This is done by increasing or decreasing the allele count by 1 in some positions. That is, each SNP will with a probability $p_{flip}$ have its value changed. The flips are always done in just one allele, so the possible flips are $0 \rightarrow 1, 2 \rightarrow 1, 1 \rightarrow 0$, and $1 \rightarrow 2$, with the heterozygote flipping to either homozygote with equal probability. This SNP flipping augmentation will change the genotype to a larger extent than masking. Introducing it makes the model resilient to individual genotyping errors. From applications such as imaging, so-called salt-and-pepper noise can frequently be used even when that specific noise structure is not expected to be present in images, as a way to combat overfitting. The overall short-range LD background should still be possible to identify even if a single variant is flipped.

When creating augmented versions of a SNP vector, we first apply the marker flip transformation, then mask, and finally, we apply a one-hot encoding to the augmented SNP vector before feeding it to the first layer of the neural network. For each genotype, the one-hot encoding takes the $n_{markers} \times 1$ dimensional vectors and maps them to $n_{markers} \times 4$ vectors. It does this by, for each SNP, mapping $0 \rightarrow (1, 0, 0, 0), 1 \rightarrow (0, 1, 0, 0), 2 \rightarrow (0, 0, 1, 0), 1 \rightarrow$ and missing values $-1 \rightarrow (0, 0, 0, 1)$. An illustration of the data augmentation strategy is shown in Fig. 3. Each sample will be augmented anew each epoch, which means that the network will most likely never see the exact same sample twice during training,

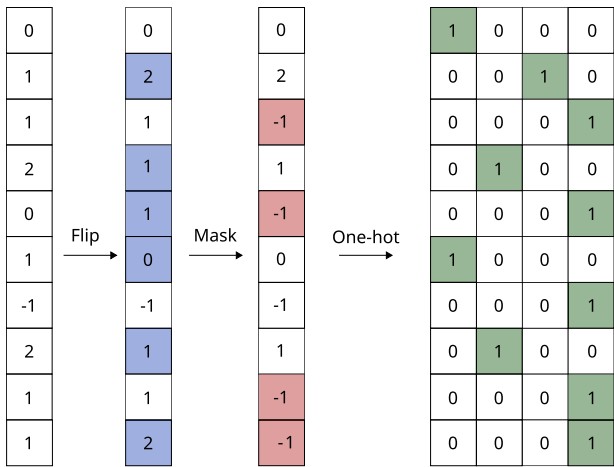

**Fig. 3.** Illustration of the augmentation strategy which consists of first flipping some variants with probability $p_{vflip}$, then masking with probability $p_{mask}$. Shown is an example of the augmentation of one sample. The shading in columns 2 and 3 indicates the markers where flipping and masking have been applied, respectively, and the final output after the one-hot encoding in the last 4 columns which is the input to the model. For illustration purposes, we show a case where both flipping and masking rates are relatively high. This procedure is done for all samples and redone every batch.

since it will be given new augmentations each time. Practically, the augmentation is implemented as the first layer of the neural network model for efficient GPU usage, and the maximum flipping and masking rates $p_{flip\_max}$ and $p_{mask\_max}$ are given as parameters to this first layer. In each training iteration, the proportions of markers to be masked and flipped for each sample are drawn from uniform distributions $p_{flip} \sim \mathcal{U}(0.001, p_{flip\_max})$ and $p_{mask} \sim \mathcal{U}(0.001, p_{mask\_max})$.

### Choice of negatives
We have investigated several options and have seen that choosing samples randomly but drawn using weights according to the inverse embedding distance in each training step to the anchor has yielded the best results.

### Model architecture
The model used consists of 2 convolutional layers with 5 filters, a kernel size of 3, and a stride of one. Then after flattening the output from the convolution, follows 3 dense layers with 256 units, with batch normalization. The sigmoid linear unit (SiLU, also known as Swish) activation function is used after each weight layer, and is defined as $\sigma(x) = \frac{x}{1+\exp(-x)}$. SiLU is essentially a smoothed version of the commonly used ReLU activation function. The model ends with a dense layer with 3 units, and an L2-normalization, resulting in samples getting mapped to the unit sphere in 3D. Figure 4 illustrates the model architecture.

In comparison to an embedding that maps individuals to a regular 2D surface, samples will now always have neighbors to compare to in all directions. The risk of a cluster, or a population, being regarded as an outlier group or genetically distinct from other samples is decreased. Visualizing a spherical surface in two dimensions is a well-studied problem, driven by the need to visualize the world map. Such a visualization cannot retain all aspects of the 3D map, and some properties will be sacrificed. We use the Equal Earth map projection (Šavrič *et al.* 2019). It is a projection that retains relative areas but does introduce some shape distortions, mainly at the poles. Supplementary Figure S1

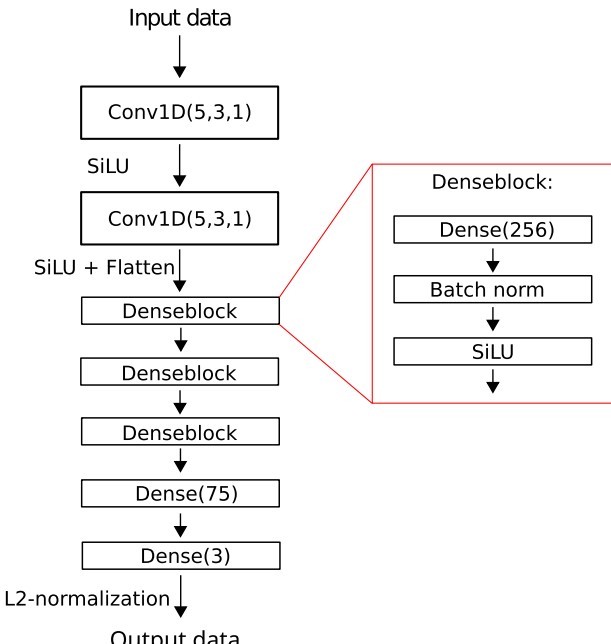

**Fig. 4.** Encoder model architecture used in the experiments, which transforms the data from the high-dimensional SNP vector, down to a 3D embedding on the unit sphere.

illustrates the transformation from 3D to 2D. The projection needs to be made with respect to a coordinate system in which the sphere is defined. The sphere can be rotated arbitrarily before applying the map projection, and how this affects the embedding and the corresponding classification performance can be seen in Supplementary Fig. S2.

Some datasets might have a population from which others have originated, and thus the population structure could be considered to have an origin, which the model should be able to capture. However, we do not want the model to assume this a priori.

Using convolutional layers is motivated by Ausmees and Nettelblad (2022) to enable the model to catch local patterns over the input. This is in contrast to t-SNE and PCA which consider each SNP location in isolation and thus lack knowledge about the relative SNP position in the genome, which we know is important information. More intricate model choices have been explored, including residual and locally connected layers, but with model complexity came increased training difficulties. A more thorough model optimization is left for future study.

For all experiments shown, the model has been trained using the Adam optimizer (Kingma and Ba 2014) with a learning rate of 0.001, $\beta_1 = 0.9$, $\beta_2 = 0.999$, with the learning rate decaying by a factor of 0.99 every 10 iterations, and trained for 5,000 epochs. The masking and flipping rates $p_{mask}$ and $p_{flip}$ are drawn anew for each training iteration, and are unique for each sample. They are both drawn from a uniform distribution between 0 and 0.99, $\mathcal{U}(0.01, 0.99)$, meaning we have a relatively aggressive augmentation strategy. As negatives for each anchor, we use all of the other samples.

## Results
### Datasets
We will focus on two different datasets in this study. A dataset consisting of 1,355 samples of 161 different dog breeds, grouped into 23 clades. Each sample has 150,067 SNPs (Parker *et al.* 2017).

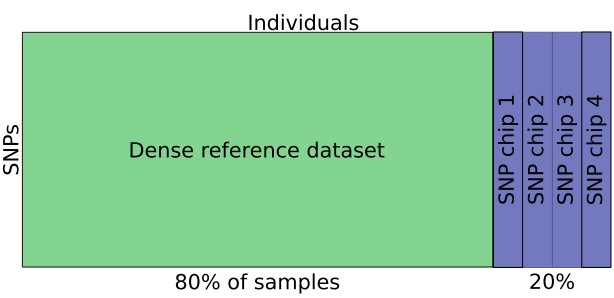

**Fig. 5.** Experimental set-up for the case where we have a dense reference set consisting of 80% unchanged samples, and the remaining 20% are split into four groups. These groups have 20% of their SNPs marked as missing, where the missingness pattern is identical for every sample in the same group. This is done to mimic a case where individuals have been genotyped by different SNP chips with lower coverage than the full genotypes.

The only filtering and preprocessing was to remove sex chromosomes, resulting in the samples having 146,330 SNPs. The variants are coded as 0, 1, and 2 for homozygote reference, heterozygote, and homozygote alternate, respectively.

We also consider the Human Origins dataset (Lazaridis *et al.* 2016), and filtered in the way described in Ausmees and Nettelblad (2022), which consists of removing sex chromosomes, noninformative sites, and applying a MAF filter of 1%. This gives a final dataset with 2,067 samples and 160,858 variants.

We are training the model using augmentations based on missing data, implicitly training the model to be robust to discrepancies or patterns in the dataset related to how the data was acquired. We devised an experiment to investigate this. Say that we have a densely genotyped dataset, and a subset that has been genotyped by more affordable SNP chips with a smaller marker set. Ideally, we want to utilize the many SNPs available for most of the samples, while not capturing patterns associated only with the SNP-chip used.

To simulate this we look at the Human Origins dataset, keep 80% of the samples as-is, and divide the remaining 20% of samples into four groups. These groups each have a unique collection of markers that will be set to missing values for every sample within the same group. In this experiment, 20% of the total marker count is masked, and chosen randomly for each group. This means that the four groups will have the same missingness pattern, mimicking a case where the same SNP-chip has been used to genotype samples in the same group. We would want these samples to be mapped according to their genetic makeup, not their missingness pattern. The idea is illustrated in Fig. 5. The missing samples were then set to the most common SNP in each position, which means that the samples from the different SNP-chips will have some common structure, not necessarily related to the population structure. This may seem like an odd and naive choice, but we saw in our experiments (not shown) that it is better than leaving them encoded as a 9, or −1, which are common representations.

All methods are trained on 80% of the individuals in both the dog and Human Origins datasets, and the other 20% are left as validation individuals to test the generalization performance. In the case of the masked dataset, the validation samples are the ones having their genotypes masked to mimic the case where we train on a dense reference set, and evaluate on new data.

Here, we evaluate our methods on empirical data. In the appendix Supplementary Fig. S5, we compare the methods on simple cases on simulated data, as well.

## Performance metrics

As discussed above, the goal is for these types of models to understand the genomic structure of the individuals and produce an informative embedding that can be used as a tool for other analyses. Some studies evaluate embedding performance by measuring the correlation between a distance in genotype space and a distance in embedding space, but we argue that this is not a good proxy for how well the embedding will perform in downstream tasks. Many traits do not necessarily have linear relationships to genotypes, and the genome has a highly correlated linkage structure which fails to be described by simple distance metrics.

Studies have also shown that at least PCA embeddings are sensitive to distribution imbalances of the dataset, and that the relative distances between population clusters are affected by the number of individuals in the different populations (McVean 2009).

The main approach we take to measure the performance of the methods is to evaluate the dimensionality reduction based on some downstream task using the embedding coordinates. In this study, we evaluate the embeddings by their population classification performance. Since we would assume that individuals from the same populations would be genetically similar, the performance of a trained population classifier should give an implicit measure of how much of the information has been retained when reducing the dimensionality of the input data.

One such classifier is the $k$-nearest neighbor (KNN) classifier, classifying a sample as the majority vote of its classes of the $k$ closest samples in the embedding. The predicted class $y_i$ for sample $i$ is computed as

$$y_i = \operatorname{argmax}\left(\sum_{j \in \mathcal{N}_i^k} \tilde{y}_j\right), \tag{7}$$

where $\tilde{y}$ is a one-hot encoding of the data label, and $\mathcal{N}_i^k$ is the set of $k$ closest samples in the embedding space. By varying the value of $k$, we can see how well the methods perform on different scales. When evaluating performance, we report the classification accuracy, that is, the fraction of correctly classified samples using the KNN classifier.

The contemporary dimensionality reduction methods we consider in this study are known to emphasize different characteristics of the data. For example, PCA tends to capture global structure and retain relative distance relationships, while having a harder time discerning local structure. The neighbor-based methods UMAP and t-SNE distinguish and emphasize local relationships in the data, while often not reflecting the global structure well. We have developed our method to capture structure on both scales, while learning a generalizable representation of the genotype data that can be applied to unseen data.

With this in mind, we seek to quantify the performance of the methods in regards to how well they preserve both local and global structure, and how well they generalize to new data. We will use the KNN classifier as a basis for most of the metrics we employ to evaluate performance but also use two metrics without label information.

### Label-based metrics

#### Local structure performance

The first way we quantify the local structure preservation of the embeddings is by using the KNN classifier to predict the subpopulation labels using the three nearest neighbors (3NN) on the training data. This relies on the label information on either the breed of the dog or the subpopulation of the individual in the Human Origins dataset, both of which should be characterized by high genetic similarity within their subclasses. The majority of an individual's closest neighbors being from the same subpopulation should indicate that the embedding preserves local structure well, and the performance is quantified by the classification accuracy of the 3NN classifier. In the rest of the text this metric is denoted by (L).

#### Global structure performance

We measure the preservation of global structure in two ways. First, we explore the evolution of KNN classification accuracy for superpopulation labels with an increasing number of neighbors $k$. We want an embedding that is relatively stable in this metric as $k$ increases, as this will capture how well the methods have separated the clusters based on superpopulation.

The second way we measure global performance is by using the two-tiered label information we have in both datasets. The dog dataset has information on the breed of every individual, and clade information for each breed. Similarly, we have information on the sub- and superpopulation labels for the Human Origins dataset. We compute the mean embedding coordinates of the samples in all subpopulations, and then use these coordinates in a 3NN classification model predicting the superpopulation. This would give a low score to a method that perfectly isolates subpopulation clusters (signified by a high (L) score) but fails to place them close to the other subpopulations that belong to the same superpopulation. This metric is denoted by (G) in the text.

#### Generalization performance

We are interested in having a dimensionality reduction model that does not only perform well on the training data but can also generalize and perform well on data it has not seen during training. To evaluate this, we use the 3NN classification model to predict the label of projected validation individuals using the labels of the individuals in the training data, and report the accuracy on both the dog and Human Origins datasets. Another generalization property that we are interested in is how well the methods perform in the experiment with artificially masked human data. With these two metrics, we measure generalization performance with regard to both unseen and incomplete data. The generalization metrics are denoted by (GE).

### Label-agnostic metrics

For a second way of quantifying the preservation of the local structure, we take inspiration from Chari and Pachter (2023) for a metric that does not use any label information. Instead of relying on labels that imply genetic similarity, this approach directly measures genetic similarity in the genotype space. For all pairs of individuals, we compute the Manhattan ($\mathcal{L}_1$) distance in the genotype space and the Euclidean distances in the embedding space. We then compute the $k$ nearest neighbors in both spaces and compute the overlap in the two sets. A high overlap in the neighbor sets indicates that individuals with high genetic similarity also get mapped close in the embedding. We measure the overlap of the $k$ closest neighbors in genotype and embedding space, for values ranging from $k = 3$ to $k = 100$, and report the mean ratio of overlap in the neighbor sets over all individuals. Results for this metric applied to the dog dataset, and we denote it by the neighbor overlap score.

Computing the overlap in neighbors provides one way to assess the consensus between two neighbor sets and quantify how well the embeddings locally reflect the structure in genotype space.

However, this approach only captures the ratio of overlapping neighbors and does not account for potential larger discrepancies in the ranking between genotype space and embedding space. To introduce an additional label-agnostic metric that considers the global structure, we evaluate an individual's $n$th closest neighbor in genotype space and compare this to its rank in the embedding space. Specifically, for a fixed value of $n$, we compute the root mean square error (RMSE) between the rank in genotype space $R_G$ and the rank in embedding space $R_E$ over all the samples as

$$\text{Rank}_{\text{RMSE}} = \sqrt{\frac{1}{N}\sum_{i \in N}\left(R_G^i - R_E^i\right)^2}. \tag{8}$$

We compute this error for values of $n$ ranging from 1 to 100 for the Human Origins dataset, and is denoted the neighbor rank RMSE score.

## Compared methods

We compare the performance of our centroid loss embeddings with PCA, UMAP, t-SNE, and popvae. We also compare to our contrastive learning framework using the same augmentation strategy and model, but with the triplet loss.

For popvae, we ran the model with the default values for the model architecture of 6 dense layers with 128 nodes. We set it to run for a maximum of 5,000 epochs with a patience of 500. For the Human Origins dataset, popvae had issues with diverging loss values, which was resolved by decreasing the initial learning rate to $10^{-4}$ which might make the training slower, but should not negatively affect the embedding quality.

The most influential hyperparameters for UMAP and t-SNE when it comes to the preservation of local and global structure are $n\_neighbors$ and perplexity, respectively, which both relate to the number of closest samples to be considered when constructing the neighbor graphs. To show the effect these parameters have on the embeddings, we show results for two values, 3 and 30, for both methods. In the coming tables and plots, the number after UMAP and t-SNE denotes the number of neighbors considered. Other than that, the default parameters are used for both implementations.

Following contemporary common practice, the t-SNE and UMAP embeddings are produced by first applying so-called PCA-preprocessing (Chari and Pachter 2023). First, the genotype datasets are reduced to 100 dimensions using PCA, and then t-SNE and UMAP are applied to the reduced datasets. Preliminary experiments showed that the PCA-preprocessing made significant improvements in both the local (L) and generalization (GE) scores of UMAP on the Human Origins dataset as compared to UMAP directly on the genotype data, while not having a major impact on the other t-SNE and UMAP results (not shown).

## Dog dataset and population classification over different scales

We perform dimensionality reduction using our contrastive learning framework on the dog dataset and compare it with other contemporary methods. Figure 6 shows the resulting embeddings using PCA, popvae, t-SNE, UMAP, and contrastive learning using the triplet loss as defined in Equation (1) and the centroid loss as in Equation (6). We show the t-SNE and UMAP embeddings with the best validation accuracy, which was t-SNE 30 and UMAP 3.

Table 1 shows the global (G), local (L), and generalization (GE) scores for the embeddings of the dog dataset, where all methods have been trained on the same training set of 80% of the individuals, and projected the last 20% to use as a validation set.

Figure 7 shows the local structure preservation of the embeddings measured by neighbor overlap score. We see a peak in all methods at around $k = 8$. This can be compared with the average number of individuals within each breed being 7.6. Before that peak, the closest neighbors in genotype space are most likely individuals of the same breed. On this scale, UMAP and t-SNE have the highest score, followed by our centroid embedding. As $k$ increases the closest neighbors will probably include individuals of another breed, but from the same clade. Here UMAP 3 and t-SNE 3 both decrease the fastest, and the neural network embeddings seem to be more robust. Figure 8 shows how the accuracy of a KNN model predicting the superpopulation vary depending on how many neighbors the KNN model considers. Notably, the neighbor-based methods have high accuracies when few neighbors are considered, but drop off quickly as $k$ increases.

Here, we have used the default UMAP implementation, but UMAP can produce spherical embeddings by changing to training using the Haversine distance. Our experiments showed however that this resulted in worse performance for all metrics as compared to 2D UMAP for the dog dataset. The performance metrics for spherical UMAP are compared with 2D UMAP in Supplementary Table S1. Supplementary Fig. S3 shows the resulting embeddings of 3D PCA and t-SNE mapped to 2 dimensions using the Equal Earth projection, and Supplementary Table S2 shows the classification performance.

## Augmentation ablation

We perform an ablation study to study the effect the two augmentation strategies have, by setting the maximum masking and flipping rates to [0.0, 0.5, 0.99], and test the different combinations. To evaluate this, we train our model on the dog dataset, and compare the 3NN clade classification accuracy on validation individuals not seen during training. The results of this experiment are presented in Table 2. The results show that with no augmentation, we unsurprisingly get bad results. Taking each augmentation in isolation, there is a clear advantage in the flipping strategy which might be surprising. We get the best results for combinations of nonzero rates, with perhaps a slight indication that a more aggressive augmentation strategy is beneficial.

## Human data visualization and built-in robustness to missing data

The label-based global (G), local (L), and generalization scores (GE) are shown for the Human Origins dataset and the masked version in Table 1. For the masked case, the main point is that the generalization score (GE) is now computed using the same unseen samples as in the Human rows, but when they have had their genotypes masked as described above. This means that the difference in (GE) score between the masked and unmasked case indicates how robust the model is to incomplete data. We see that most models suffer a substantial decrease in performance, while our centroid embedding retains comparable performance.

Figure 9 compares the embeddings using popvae, t-SNE 30, and our centroid embedding on the masked human data experiment. The left and right plots show the same embeddings, but the left plots are labeled by population, and the right plots by the missingness pattern where the points of the same color have had the same SNP masked.

The neighbor rank RMSE metric for the embeddings of the Human Origins dataset is shown in Fig. 10. The results show that

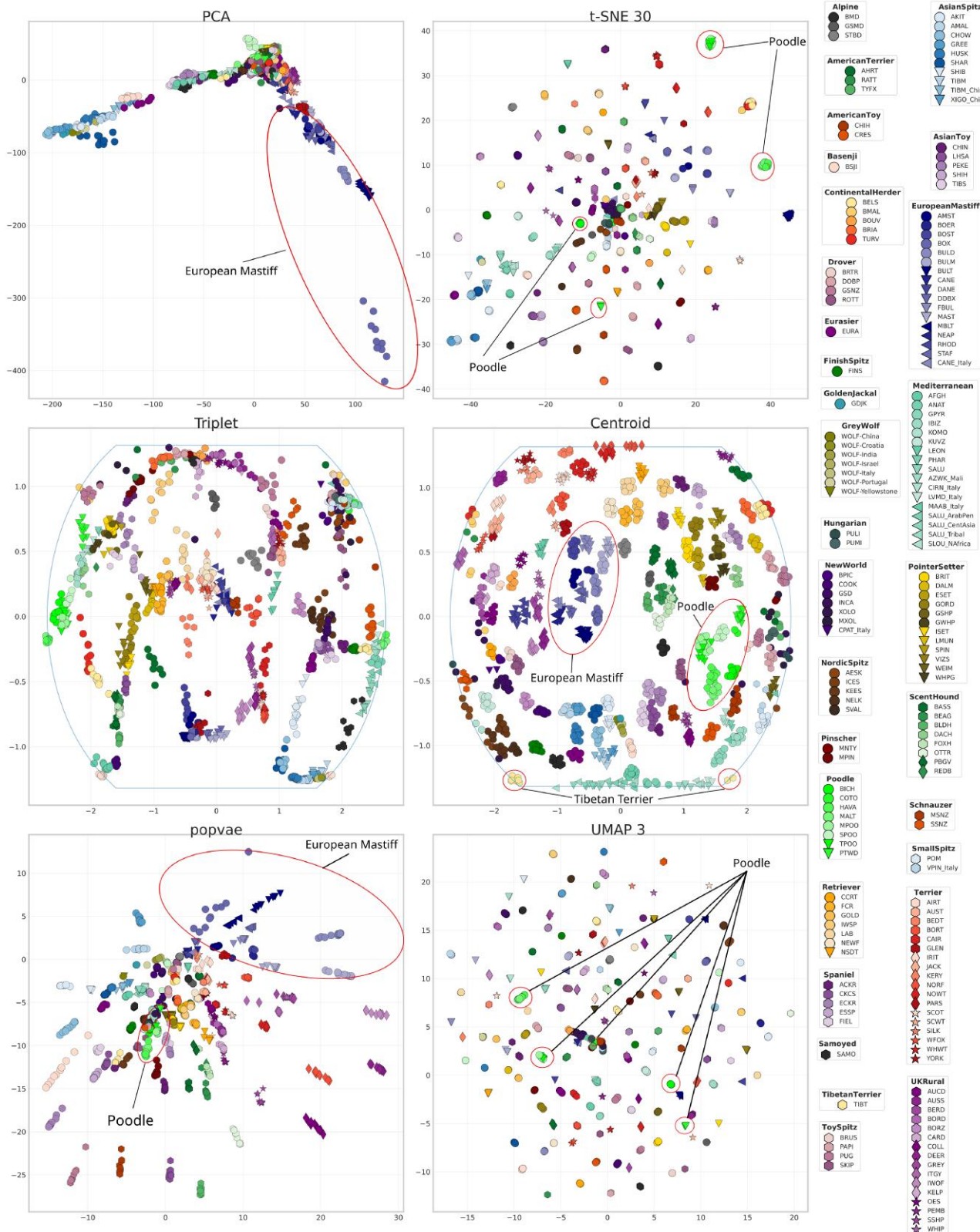

**Fig. 6.** Dimensionality reduction results on the dog dataset using PCA, t-SNE, and contrastive learning using the triplet loss, and our centroid-based loss. The legend is shown to the right, which also groups individual breeds into clades. The line in the triplet and centroid plots enveloping all samples indicates the boundary of the map projection of the spherical embedding. Clades discussed in the discussion are highlighted in some of the embeddings.

in terms of recreating approximate embedding neighbor rankings in the genotype space, PCA is the best, followed by our centroid embedding, and then popvae.

Apart from the quantitative differences in the generalization score (GE) for the masked data there are some qualitative differences between how the masked samples are projected. The

**Table 1.** The global structure (G), local structure (L), and generalization (GE) scores of the different methods based on KNN classifiers with $k = 3$ on the embedding coordinates for the three datasets.

| Method | Dataset | Superpop clustering (G) | Subpop acc. (L) | Validation (GE) |
|---|---|---|---|---|
| Centroid | Dog | **0.6723** | 0.9327 | **0.9225** |
| Triplet | Dog | 0.5141 | 0.7223 | 0.7749 |
| Popvae | Dog | 0.5763 | 0.8708 | 0.6863 |
| PCA | Dog | 0.4011 | 0.3635 | 0.3173 |
| t-SNE 3 | Dog | 0.2599 | **0.9345** | 0.8524 |
| t-SNE 30 | Dog | 0.5480 | 0.9271 | 0.9041 |
| UMAP 3 | Dog | 0.3107 | 0.9207 | 0.9114 |
| UMAP 30 | Dog | 0.4294 | 0.9022 | 0.8229 |
| **Method** | **Dataset** | **Superpop clustering (G)** | **Subpop acc. (L)** | **Validation (GE)** |
| Centroid | Human | 0.8012 | 0.5953 | 0.4638 |
| Popvae | Human | **0.8373** | 0.6999 | 0.4855 |
| PCA | Human | 0.7349 | 0.4386 | 0.4300 |
| t-SNE 3 | Human | 0.6145 | 0.7610 | **0.6111** |
| t-SNE 30 | Human | 0.7831 | **0.7701** | 0.5990 |
| UMAP 3 | Human | 0.4759 | 0.7253 | 0.5894 |
| UMAP 30 | Human | 0.7349 | 0.6467 | 0.5097 |
| **Method** | **Dataset** | **Superpop clustering (G)** | **Subpop acc. (L)** | **Validation (GE)** |
| Centroid | Human masked | 0.8072 | 0.5983 | **0.4614** |
| Popvae | Human masked | **0.8133** | 0.6461 | 0.2101 |
| PCA | Human masked | 0.7169 | 0.4392 | 0.0870 |
| t-SNE 3 | Human masked | 0.5783 | 0.7520 | 0.2609 |
| t-SNE 30 | Human masked | 0.7771 | **0.7623** | 0.3092 |
| UMAP 3 | Human masked | 0.4578 | 0.7090 | 0.2826 |
| UMAP 30 | Human masked | 0.7530 | 0.6534 | 0.2585 |

Using an 80/20 train/test split for all methods both in training the embeddings and the KNN classifier. Top scores for each dataset and metric is highlighted in bold.

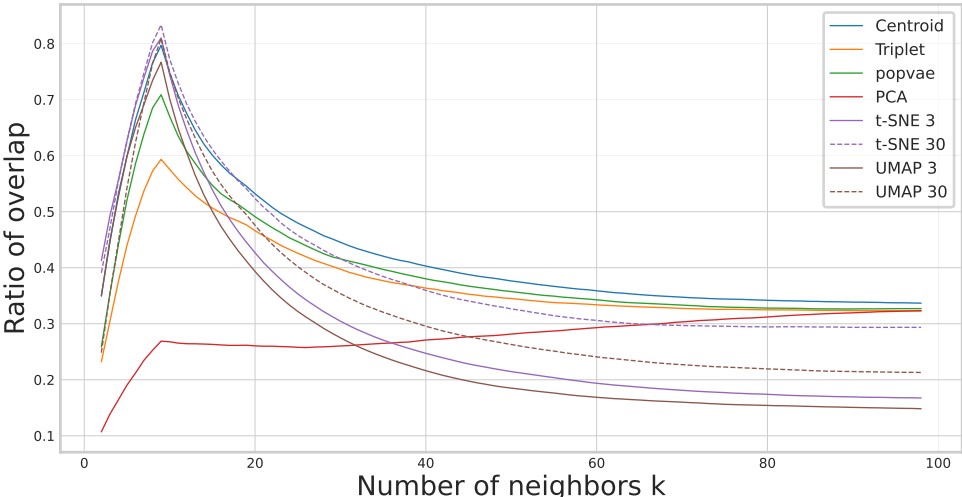

**Fig. 7.** The local structure preservation metric inspired by Chari and Pachter (2023) computed for the embeddings created on the dog data which measures the overlap of the $k$ nearest neighbors in embedding space and in genotype space. This metric does not utilize label information but instead defines genetic similarity by the $\mathcal{L}_1$ distance between genotypes.

contrastive centroid embedding generally seems to place them inside clusters of training samples (in blue), and are dispersed within them. The t-SNE 30 embedding also places the validation individuals inside the clusters, but embeds validation samples close to each other. For popvae, the validation samples have a more reasonable spread and are not clumped like in t-SNE. However, there is a clear tendency for the masked samples to be mapped closer to the origin than the training samples. The popvae embedding has a clear center of gravity that is also seen for the dog dataset, which has an apparent effect on how the masked samples are projected. This effect is absent in our contrastive embedding.

## Discussion

Figure 6 shows that PCA, t-SNE, UMAP, and especially popvae have samples radiating from an origin. This visual cue can enable interpretations that misconstrue the true population structure. This effect could also influence any quantitative analysis based on the embeddings. If we have an imbalanced dataset in terms of the number of samples from different populations, samples from less represented populations will tend to be embedded further from the origin than the populations with heavier representation. The coordinates in an embedding will depend on many factors and while sometimes

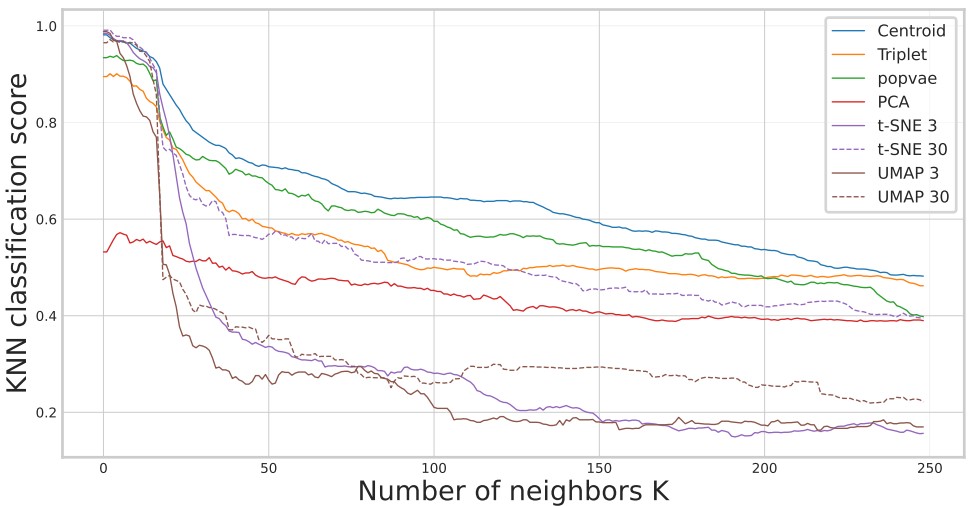

**Fig. 8.** The superpopulation classification score using a KNN classifier on projections of the dog data using PCA, t-SNE, UMAP, popvae, and our contrastive learning implementation. We are varying k on the x-axis, showing the trends in model performance on different scales. The score is computed over all samples.

**Table 2.** Ablation study over different combinations of hyperparameters in the augmentation scheme.

| $p_{flip}$ | $p_{mask}$ | | |
|---|---|---|---|
| | **0.0** | **0.5** | **0.99** |
| 0.0 | 0.239852 | 0.839852 | 0.822878 |
| 0.5 | 0.957196 | 0.956458 | 0.967528 |
| 0.99 | 0.956458 | 0.972694 | 0.971218 |

The score presented is the 3NN accuracy on the held-out validation individuals in the dog dataset when predicting the clade label.

some populations might have diverged at an early point from some others, measuring distances in any embedding can at best serve as a source for such a hypothesis. However, methods with a clear center of gravity might lead readers to these types of conclusions when some populations are embedded far from the origin, or center of gravity of all samples. Our method enables the embedding to lie more on a continuum—there is no notion of an origin, and each sample will have neighbors in all directions. The continuum property comes from the spherical embedding. Topologically it can be seen as a connected 2D surface, with a wrapping property. Going from the sphere to 2D using the map projection might introduce some artifacts. For example, in the centroid embedding in Fig. 6, the Tibetan Terrier clade has been split in the bottom right and left corners of the embedding. It might look like they are embedded far apart, but with the knowledge that the embedding is spherical, we can see that they indeed are neighbors in the embedding space. We have made attempts at making 2D embeddings confined to a box that wraps around both the x- and y-directions to force samples to have neighbors in all directions but found the spherical embeddings to be more natural and easier to implement. The lack of an origin in our embeddings comes from the centroid loss by design, as it compares the samples in a local reference frame, not affected *where* in the embedding space the individuals are, but only how they are placed in relation to each other.

For example, in Fig. 6 both PCA and popvae embed subpopulations from the European Mastiff clade (dark blue) at a large distance from the origin, well distinguished from other clades. This opens up for invalid interpretations of their status within the

study population. They should not be viewed as a peculiar breed of dog, based on this data alone. Our embedding does not have this property, where the European Mastiffs are embedded close together, both within breed and clade, somewhere on the sphere. Depending on how the spherical coordinate system is defined, its origin will dictate which samples will lie on the boundary of the embedding after applying the map projection.

The metrics presented in Table 1 show that for the dog data, our centroid embedding is the best of all methods at reflecting the global structure (G), followed by popvae. This can also be seen in the plots in Fig. 6. As an example of how this score is reflected in the visualizations, we can look at the Poodle clade (neon green). Both t-SNE and UMAP group them well within breeds, which is also generally reflected in the high local 3NN scores (L), but the different breeds that form the poodle clade do not constitute a contiguous cluster in the embedding space. In both the centroid and popvae embeddings, we see relatively good clustering at the breed level, and they are also embedded close to other poodle breeds.

Figure 8 shows that, while t-SNE and UMAP beat our method in peak classification accuracy using a KNN classifier for low values of k, the contrastive learning model performs better when considering more neighbors in superpopulation classification. This measures how well the methods embed not just the few closest related samples similarly, but how it captures the overall population structure as defined by the clade label information. This result is corroborated by Fig. 7, which shows that the neural network models are more robust in reflecting the local structure as defined by distances in genotype space when the number of neighbors increases. This indicates the commonly observed good local performance of the neighbor-based methods, and their inability to represent the global structure.

The results in Fig. 10 showed that PCA and the neural network models provide more stable neighbor rankings, signified by a lower RMSE compared to UMAP and t-SNE. This result can be viewed in combination with the overlap score in Fig. 7, where PCA performed badly. A joint interpretation of these two outcomes is that while not always reproducing the full closest neighbor set, PCA and neural network embedding rankings are never far off. UMAP and t-SNE, on the other hand, produce good average overlap scores, but this is done by sacrificing

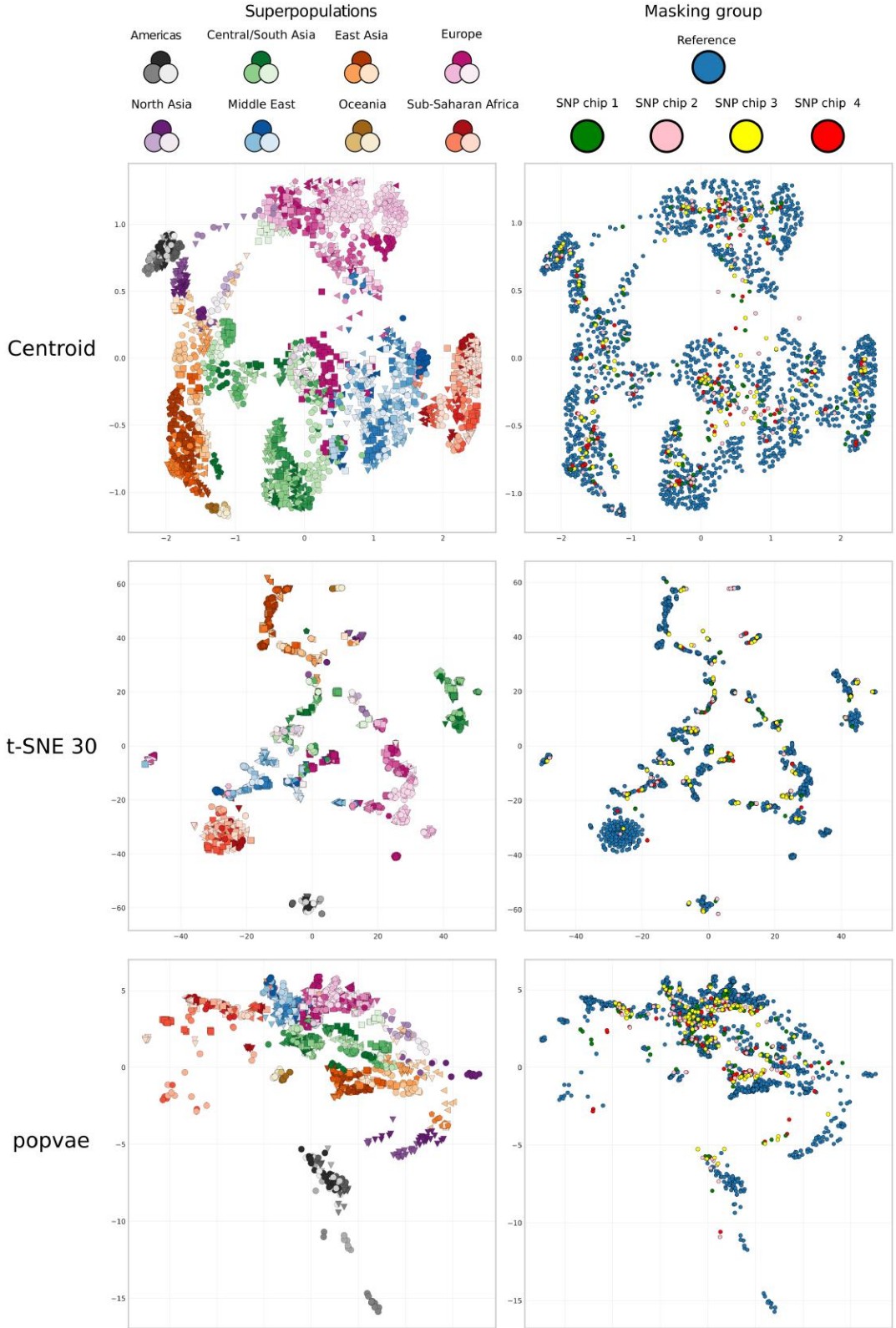

**Fig. 9.** Dimensionality reduction results comparing our centroid method, t-SNE 30, and popvae for the synthetically masked case for the Human Origins dataset. Left figures are labeled by population label, where sub-population markers are grouped by superpopulation according to the legend at the top. The right plots are labeled by the masking group, with training individuals being the majority group, and the respective masking groups indicated according to the legend above the plots.

some parts of the neighborhood structure, where some neighbors can instead be located in a completely different region of the embedded space.

Comparing the neighbor-based methods when using a different number of neighbors, from the results in Table 1 we see that when increasing the number of neighbors from 3 to 30 the global

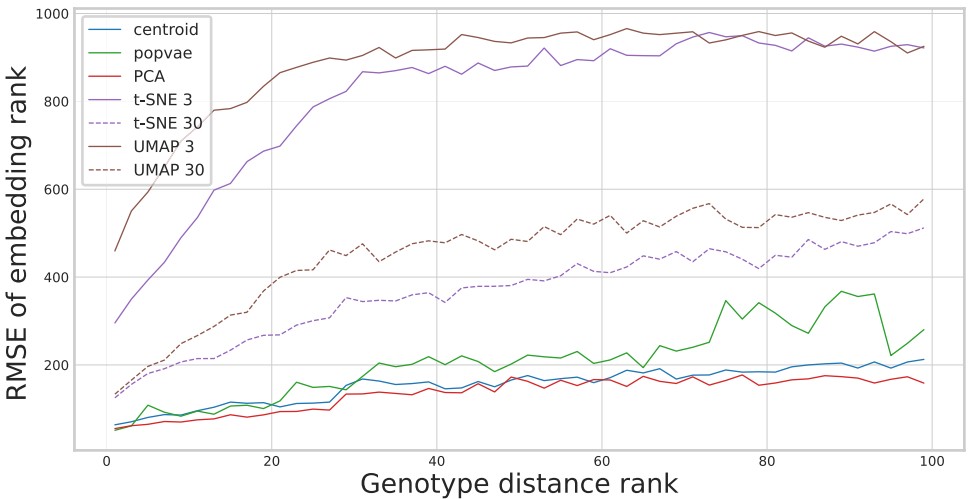

**Fig. 10.** RMSE between the rank of a neighbor in genotype space and the rank of the corresponding individual in the embedding space (low values are better), computed on the Human Origins dataset. PCA and the neural network model have the lowest values, performing the best. UMAP and t-SNE have higher values, meaning that individuals close in genotype space sometimes get embedded far away from each other.

structure score (G) increases. This is expected, but at least for UMAP, this change affects the 3NN classification accuracy both on training samples (L) and validation samples (GE) negatively. While not achieving as high local scores (L) as the neighbor-based models, the neural network models consistently outperform them in terms of preservation of global structure (G).

The augmentation strategy chosen should ideally be adapted to the dataset and the target application at hand. We ideally want to have augmentations that yield new individuals similar to the original genotypes, with respect to properties we want to retain. Here, we have chosen to create new versions of individuals by both masking some of the SNPs, and changing the value of some of them. We chose to use the masking strategy since it only hides parts of the genotype, and the augmented sample can be seen as a lower-quality representation of the exact same individual. Training a model to be robust to the masking augmentation allows it to represent samples that have been genotyped at a lower density using a different SNP chip. The ablation study shows that the marker flipping strategy is a good augmentation for the model to discern population structure in this application. Other augmentation strategies may be better suited for applications not targeting population history. For example, marker flipping might not be suitable for applications where we are interested in investigating marker effects.

Using the KNN population label classifier is an attempt at quantifying the performance of the embeddings. This choice is not necessarily the best or only way to evaluate the methods. It is chosen because of the intuition that we would think individuals from the same population to be genetically similar and thus be embedded close to each other. This is however not necessarily true, since it does not consider the relative embedding distances and how it relates to the differences in genetic make-up. There exists no ground truth that we know should hold and can compare to. PCA has often been regarded to have good, linear properties, and the resulting embeddings can reflect the geographical origin for natural populations. Recent studies show that PCA can be biased when visualizing population structure, especially when the dataset has a skewed population label distribution, with some highly represented with many samples, and some less represented with fewer samples (Elhaik 2022).

We observe good generalization properties in two ways. First, in terms of classification accuracy on unseen samples, our centroid

embedding achieved the highest validation score (GE) on the dog dataset, and while it performed worse than popvae in terms of local performance (L) on the training set of the Human Origins data, it still performs on the same level on the validation data (GE), and in terms of capturing global structure (G). Furthermore, the neural network model validation accuracy is not much lower than the training accuracy, which suggests that the model does not simply remember the samples it has seen, but can learn more abstract properties mapping similar samples to similar regions on the sphere. The other generalization property is showcased in the experiment on the masked human data. Figure 9 shows that having introduced an artificial but still domain-relevant pattern in the data, t-SNE clusters based on population label but also based on the missingness pattern. This is an undesired artifact in a population structure study. We would want to incorporate as much information available to us as possible, but t-SNE is not able to make proper use of this data. In contrast, our method and popvae seems more robust to this kind of perturbation in the data and does not visibly cluster on the missingness pattern. Both generalization properties are most likely observed due to the comprehensive data augmentation scheme used in our contrastive learning implementation.

The runtime for the dog dataset on one A100 Nvidia GPU is around 1.3 s per epoch, or around 107 min for the full 5,000 epochs, while it takes less than a minute for the three methods PCA, t-SNE, and umap. The runtimes for popvae was around 0.8 seconds per epoch and finished early after 773 epochs for a full run time of 10.9 min. One reason for this difference in times despite having similarly sized encoder models is that contrastive training requires the model to compute the embeddings for two views of each sample in every batch. This means that we currently are not competitive when it comes to computational cost. For this reason, we have implemented a distributed framework that can utilize multiple GPUs to process batches in parallel. This means that we can speed up the wall clock time for training significantly given more resources.

Having an explicit mapping makes drawing conclusions about how the input affects the output more tractable. An increasing number of applications rely on deep learning (DL) to achieve previously unreachable levels of performance. However, the neural network models are considered black boxes, where the user is blind to the inner workings of the machinery behind predictions. In some applications, one may be content with getting good performance, at

the cost of not being able to explain why the conclusion was drawn. In other applications, the interpretability of outputs is important, for example, in disease detection and computer-aided diagnosis (Meng *et al.* 2022). Also, in association studies, we want to be able to say something about the feature importance in terms of which variants contribute to desired traits, information that could improve future breeding decisions. Explainable AI (XAI) is an emerging field, where the aim is to make the black boxes more transparent (Xu *et al.* 2019). One example of such an approach is salience maps for image classification models, where the methods can indicate which regions of the image had the most influence on the class prediction (Simonyan *et al.* 2014). There is an interesting parallel that could be explored, where one could use similar methods to explore which variants contribute to phenotype predictions using neural networks.

While we are not there yet, we hope that this study is a step towards creating larger neural network models, with more generalized higher-dimensional embeddings that can serve as foundational models. These models could be used to build improved analysis models for many different applications—akin to a ResNet (He *et al.* 2016) of genotype data. The creation of a reference model by first training on a large dataset to learn the structure of the genotype data, can with transfer learning techniques bring great benefit when fine-tuned to much smaller study populations, and can constitute a potential remedy for the problems that usually comes with the low sample count in many genomic studies and datasets.

## Data availability

The code written for this project is openly available at https://github.com/filtho/ContrastiveLosses. This work introduces no new data, but focuses on methodological work. The data used in the examples are openly available from the corresponding studies. The Human Origins dataset from Lazaridis *et al.* (2016) can be found at https://reich.hms.harvard.edu/datasets. The dog dataset from Parker *et al.* (2017) is available at https://research.nhgri.nih.gov/dog_genome/downloads/datasets/SNP/2017-parker-data/. The software is written in Python, and TensorFlow (Abadi *et al.* 2015) version 2.15. The PCA embeddings are produced with SciKitLearn version 1.5.0 (Pedregosa *et al.* 2011). The t-SNE embeddings are computed using OpenTSNE version 1.0.2 (Poličar *et al.* 2019), and the UMAP embeddings are computed using umap-learn version 0.5.6 (McInnes *et al.* 2018).

Supplemental material available at GENETICS online.

## Acknowledgments

We thank the editor and reviewers for the detailed and valuable feedback which led to the improvement of this work.

## Funding

Compute resources provided by SNIC through the National Supercomputer Centre (NSC) at Linköping University under Project Berzelius 2022-17, 2022-165, 2023-258, and 2024-121. Project funded by Formas, The Swedish government research council for sustainable development, Grant no 2020–00712, Deep Learning for Analyzing Population Structure and Genotype-Phenotype Mapping.

## Conflicts of interest

The author(s) declare no conflicts of interest.

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
