## [Peer Review File · Genetics]

Dimensionality Reduction of Genetic Data using Contrastive Learning

Filip Thor and Carl Nettelblad

NOTE: The reviews and decision letters are unedited and appear as submitted by the reviewers.

In extremely rare instances and as determined by a Senior Editor or the EIC, portions of a review may be redacted. If a review is signed, the reviewer has agreed to no longer remain anonymous.

The review history appears in chronological order.

Review Timeline:

Submission Date:	2024-09-27
Editorial Decision:	2024-12-08
Resubmission Received:	2025-01-21
Editorial Decision:	2025-02-24
Revision Received:	2025-03-05
Accepted:	2025-03-26

December 8, 2024

GENETICS-2024-307497

Dimensionality Reduction of Genetic Data using Contrastive Learning

Dear Filip:

Two experts in the field have reviewed your manuscript, and I have read it as well. The use of contrastive learning for producing low-dimensional embeddings is innovative and the principles are well explained in the manuscript. While your manuscript is not currently acceptable for publication in GENETICS, we would re-consider a substantially revised manuscript. Both reviewers have comments and concerns to be addressed in a revised manuscript. You can read their reviews at the end of this email.

The overall assessment is that the technique is interesting but that the delineation of the advantages over previously published approaches are not well established. As the comments explain, the comparisons to other methods would be better if they were more extensive in terms of comparing to other top algorithms that have presented in the literature (popvae, UMAP), and if they explored performance on other tasks besides classifications based on the low-dimensional embedding. As it is, the manuscript is most interesting for novel methodology, but in ways that might be more suitable for a statistical learning audience than the Genetics audience.

We look forward to receiving your revised manuscript. Please let the editorial office know approximately how long you expect to need for revisions.

Upon resubmission, please include:

1. A clean version of your manuscript;
2. A marked version of your manuscript in which you highlight significant revisions carried out in response to the major points raised by the editor/reviewers (track changes is acceptable if preferred);
3. A detailed response to the editor's/reviewers' feedback and to the concerns listed above. Please reference line numbers in this response to aid the editor and reviewers.

Your paper will likely be sent back out for review.

Additionally, please ensure that your resubmission is formatted for GENETICS
<https://academic.oup.com/genetics/pages/general-instructions>

Follow this link to submit the revised manuscript: Link Not Available

Sincerely,

John Novembre
Associate Editor
GENETICS

Approved by:
Hongyu Zhao
Senior Editor
GENETICS

Reviewer #1 :

In this manuscript, authors proposed a representation learning method on genomic data that utilize the contrastive learning framework. Overall, I think this is an interesting method. There are a couple of places where I think the authors could fill gaps in the story that would make it more complete.

1. According to the manuscript, the authors use classification score as the evaluation metric for the learned representation. However, the true cluster structure of the population is unknown. and the typical downstream task in representation learning is often prediction. I suggest that the authors design a downstream task, such as genomic prediction, to better evaluate the performance of the learned representation.

2. Regarding the data augmentation, I think masking SNP data to generate positive pairs is a good idea. However, I'm unclear on

the rationale for flipping SNPs. Flipping a SNP can result in a completely different genetic meaning. It would be helpful if the authors could add a paragraph discussing this choice.

3. When generating positive pairs, how is the proportion of masked alleles controlled?

4. The novelty of this method seems to be the application of the centroid idea to the N-pair loss. I wonder if this approach could also be applied to other well-known contrastive loss functions, such as the NT-Xent loss.

5. In equation 3, where the centroid is computed, the negatives are multiplied twice. How is the weight of the negatives determined? Is this adjustable, and can it be treated as a hyperparameter?

Reviewer #2 :

Review of Thor and Nettelblad

In this paper the authors develop a framework for contrastive learning in population genetic data. Briefly, the authors utilize a data augmentation approach for calculating various contrastive loss functions in the context of a deep convolution architecture that takes as input genotype data that with a masking channel. This sort of contrastive learning is encoder only and thus yields an embedding space of the data, which the authors then describe with a 2D visualization and KNN classification metric. Comparisons are made to embeddings made with PCA and t-SNE. The results are interesting and the discussion thoughtful and well-pitched.

I have a few suggestions / queries aimed at improving the paper and to perhaps for thoroughly evaluate the methods proposed.

Bigger stuff:

- The authors evaluate the model on two datasets- one from human and one from dogs. While evaluation on empirical data is the ultimate goal, it would be helpful to also apply the contrastive learning method on simulated data. In particular multi-population simulations could be performed in the context of the coalescent where the 'ground truth' would be known more fully. A comparison of genealogical distance in the simulated ARG versus learned embedding space could be particularly interesting to compare, in addition to the metrics and visualizations the authors are already using. An easy way to accomplish this would be to use `stdpopsim`.
- In addition to the comparisons being made here, it would good to compare the learned embeddings to those from earlier autoencoder methods like GCAE or popvae. How much better is this new method? How do training costs/times compare?

Other points

- The data augmentation scheme requires some arbitrary choices. What are the effects on learning of the inclusion of flipping? What does performance look like without it? What effect does the hyperprior on the prob(missing data) have? Perhaps some simple supplemental tables can be adding showing comparisons of loss with different values of `p_flip` and `p_mask`?
- The embedding has output dimension = 3, but for visualization the authors do a 2D projection. What is the effect of changing the output dimensionality, in particular to 2? What about increasing the embedding dimensionality? Do higher dimensional embeddings lead to greater KNN accuracy? What do 2D projections of higher dimensional outputs look like?

Minor issues:

-L19P4 - typo "but be" repeated

-L30P6 - typo missing word

-L49P6 - reference to McVean out of place

In addition I have suggestions for the github repository:

- I'm a bit confused about the inclusion of the ``run_Cl.py`` script and the MNIST image examples. This seems like a separate project no? It's not clear what's going on with that section of the readme
- I didn't have lunch running the code on my system I was getting an error of the form:
``len` is not well defined for a symbolic Tensor (autoencoder_1/sep_sparse_oh_1/Shape:0). Please call `x.shape` rather than`

`len(x)` for shape information.`

I was forced to use a slightly different tensorflow version for the singularity app (tf 2.18) so perhaps this could explain the error. The nvidia container the authors are using uses an older tf version, so it would be great if they could support newer tensorflow versions in their revision.

Reviewer #3 :

I have attached a document with detailed comments.

For the data availability, please add the URLs for the data rather than referring readers to citations.

Associate Editor Comments:

1) I agree with the reviewers that the performance evaluation needs more expansion to evaluate if the methodology is sufficiently an advance over previous work to be relevant to practicing geneticists. For the classification task presented, 2D-PCA will clearly do poorly in classification settings with more than 2 populations, and tSNE is known to have disadvantages relative to popvae and UMAP - so I agree with the others that the experiment is not yet complete/satisfying.

For evaluating the performance more thoroughly - besides comparing to UMAP and popvae, you could take inspiration from the Chari and Pachter (PLos Comp Bio 2023) and carry out experiments to show how the method performs at preserving: 1) Local structure; 2) Global structure; 3) Distance in the multi-dimensional space.

2) There is a sense in the reviews that the paper is imbalanced with relatively few results for a long methods section. I did not mind the longer methods as it was explanatory and did a good job at that. Hopefully adding more results will balance the manuscript. That said, I could also see strategically taking the more didactic components of the methods section and moving them into a supp info section titled akin to "Introduction to Contrastive Learning" such that your main text can tell on the specific innovations / choices made in your team's implementation of a contrastive learning approach for dissecting population structure.

Review – Dimensionality Reduction of Genetic Data using Contrastive Learning

Reviewer description of the manuscript

In this manuscript the authors present a framework to use contrastive learning, a form of dimensionality reduction using neural networks, to visualize population structure. The manuscript provides in-depth detail on the method and tests it on two datasets (one for humans, one for dogs). In contrast to existing methods such as PCA, t-SNE, and UMAP—which use single nucleotide data—this approach uses convolutional layers which the authors state also may capture patterns of linkage. Their approach also uses a loss function defined specifically for SNP data. Computationally, the method is very slow, but can be sped up using parallelized GPUs.

General comments

- This is a small comment but the authors alternate between “NN” and “neural network” throughout the manuscript. I suggest sticking with “neural network” as later in the manuscript they discuss KNN (k-nearest neighbours) and it can get confusing with “NN”.
- In the introduction they discuss the uses of t-SNE and UMAP in visualizing population structure. They also state that the usage of these methods has “sparked controversy and faced community criticism”. This is a valid statement, however it is not clear what the controversies and criticism they are referring to. This could possibly refer to the UMAP recently presented in the All of Us flagship paper (discussed in Kozlov, 2024, Nature); alternately, it could refer to some of the critiques specific to neighbour embeddings or to the methods specifically (e.g. that they do not have interpretable long-range distances).

I believe this is important since it motivates the rest of the manuscript. If it is in response to how race, ethnicity, and genetics are presented in figures, I do not think that a new method adequately addresses the problem regardless of whether it has robust theoretical properties, *particularly* if it is motivated by population classification. If it is with respect to addressing specific methodological shortcomings of PCA, t-SNE, UMAP, etc., then the authors must be explicit about the shortcomings and how their method seeks to address them.

- Throughout the manuscript they describe one of the shortcomings of t-SNE and UMAP as methods that are unable to project new data onto an existing embedding in a straightforward manner. UMAP has long had this ability in its main Python implementation (via the **fit** and **transform** methods); the scikit-learn version of t-SNE does not have this ability, but it is available in newer packages (e.g. openTSNE). If it is a computational or theoretical issue (e.g. it is overly heuristic), the authors should clarify this point.
- The authors compare their method to t-SNE and PCA throughout the manuscript; I believe it would also be appropriate to compare to UMAP. Using their method, they project data to a three-dimensional sphere which they then flatten. While t-SNE cannot do this, UMAP has the ability to specify the type of space to which data are projected (see documentation). It would be worthwhile to make this apples-to-apples comparison, particularly since I haven’t seen others take this approach to 2D visualizations.

- Given that it is a neural network approach, I think it would be important to compare to existing methods that they cite (such as GCAE), if not as part of the analysis then at least citing its performance relative to GCAE.
- For t-SNE, it would be helpful to specify the parameters used (particularly perplexity). This could go in the supplement if necessary.
- The description of the method is quite long and comprises about half of the manuscript's text (approximately 3.5 pages of methods text vs 3.5 pages of other text). Some of this could be shortened or moved to the supplement (e.g. the portions about Model Architecture related to spherical embeddings or hyperparameters for the Adam optimizer).
- The Human Origins data is available online, but it would be nice to have a direct link to it along with the citation. The dog dataset is linked on their github repo.
- The results about how to code missing data (-1 vs 9 vs imputing the most common variant) are minor but still interesting.
- The contrast between methods in Figure 8 is also interesting and I am curious if there are any consequences to the (possibly divergent) conclusions you could draw from comparing these different clusterings. This seems like an important difference but it is not deeply explored.
- For the data availability, please add the URLs for the data rather than referring readers to citations.

Specific comments

Comments will be formatted as [Page] – [Line number] or as [Figure].

- 1 – Line 45: As mentioned above, be specific about what the controversies and criticisms are.
- 2 – Final two paragraphs: Can this method actually preserve patterns of linkage? This seems to me like an important strength that would set it apart from SNP-based methods but is only mentioned in passing as a possible theoretical benefit. This type of analysis has been done before (see e.g. Battey et al, 2021, Fig. 9)
- 2 – Line 105: “in a way an indirect way” is unclear
- Figure 1: Some typos in the last sentence. It should read “In both models, the embedding coordinates are used as the population visualization.”
- 3 – Line 5: Just to be clear, a “batch” is a group of individuals rather than a collection of SNPs, correct?
- 3 – Line 10: I would split this sentence as it is unclear. I would also recommend defining “anchor”, “positive”, and “negative” for readers unfamiliar with machine learning terminology.
- 4 – Line 75: I’m unclear on how the data are encoded. Does pflip determine the type of flip (e.g. 1 to 0 vs 1 to 2)? Or rather just that it will be flipped? Also how does the system determine the direction of a flip for the value 1? E.g. does it go 1 to 2 or 1 to 0?

- 4 – Final two paragraphs: The order of the text suggests that you first mask and flip, while the figure illustrates that you flip and then mask. I suspect the latter is the case; if so, I would re-arrange the text to be clearer. I would also be explicit about what the one-hot encoding does. It is mentioned in the text and left to the reader to infer based on Figure 3.
- Figure 3 – This figure introduces the term “sparsify”, which I believe refers to the masking. As far as I can tell this is the only usage of the word “sparsify”—it would be clearer to write “masking” (or to define “sparsify” in the text somewhere). The one-hot encoding is not obvious at a glance, though I’m inferring that $\{1,0,0,0\}$ represents unchanged data, $\{0,1,0,0\}$ represents a SNP flipped to “1”, $\{0,0,1,0\}$ is a SNP flipped to “2”, and $\{0,0,0,1\}$ is a SNP that has been masked. This explanation would be useful in the text or caption.
- Figure 4 – The flow in this figure needs to be tidied. Text next to the arrows indicates both data (“Input data”) and processes (SiLU). It should finish with “Output data” or something similar, as it currently has an arrow with “L2 normalization” next to it and points to the caption text.
- 5 – Line 10: I’m confused about how you choose samples randomly based on the inverse embeddings distance given that the embedding is generated at the end of the process. Is this done with an initial embedding before optimization, or its distance in the high-dimensional space?
- 5 – Model architecture: I believe UMAP would be a good comparison to this approach rather than t-SNE. It is fairly straightforward to project to a 3D unit sphere and I think this would be a very interesting result.
- Figure 5: Typo in “This to mimic samples”
- 6 – Performance metrics: Stray citation to McVean 2009 at the start of the section.
- 6 – 57: Sentence is confusing. I understand the point but it could be simplified to be clearer.
- 6 – 60: This sentence also needs simplification as the word “sample” is playing three different roles.
- Figure 6: It is very difficult to make out any details in the figure because there are so many combinations of shapes and colours. Consider collapsing some of the categories.
- 8 – 31: Is this a consequence of your contrastive learning approach? Or is this the result of choosing to embed your data in a 3D sphere and then flattening that into 2D?
- 8 – 37: It is not clear where I am supposed to find the points in the t-SNE embedding. If you’d like to highlight specific points, I recommend adding a new panel where the other points are grey or more faint and the points you wish to discuss are clearly highlighted. I am also not sure I agree with the interpretation... when a reader is presented with a figure, won’t some cluster always be outlying?
- 8 – 48: Why not compare in 3D? Visualization and classification are separate tasks and I would imagine that given the capability we would do classification in 3 dimensions rather than 2.
- 8 – 86: “entire dataset” repeated.
- Figure 8: Do the different shades or shapes represent anything?

- 11 – 6: I believe you mean “transparent”, not “translucent”

Review response: Dimensionality Reduction of Genetic Data using Contrastive Learning

Filip Thor, Carl Nettelblad

Overall summary

We would like to thank the reviewers for their high-quality and constructive reviews. They have been helpful in improving the manuscript with regards to clarity. We also believe that the new version provides a more nuanced description of the improvements and differences that our methods bring compared to other contemporary approaches.

We have made substantial revisions and improvements to our manuscript based on the reviews. The most major changes include a more structured way of evaluating the performance of different embeddings on local as well as global scales. We introduce additional metrics including ones based on distances in genotype space, and a more organized way of presenting them. We have also run experiments with UMAP and popvae to compare with our method. Furthermore, we now use the fit/transform functionality in OpenTSNE as well as UMAP. Thus, all methods now use the same 80% split as training samples and project the last 20%, in comparison to before where the t-SNE embeddings were produced using all individuals. This makes some of the results look different, but most conclusions still stand from the first submission.

We indicate where in the manuscript (Page P. and Line L.) we have made changes related to the points raised by the reviewers, both in the clean version and the one where we have indicated changes in the format $\underbrace{[P. aa, L. bb]}_{Clean} | \underbrace{[P. cc, L. dd]}_{Changes}$.

Review responses

1. **Reviewer 1:** *In this manuscript, authors proposed a representation learning method on genomic data that utilize the contrastive learning framework. Overall, I think this is an interesting method. There are a couple of places where I think the authors could fill gaps in the story that would make it more complete.*

1.1. **According to the manuscript, the authors use classification score as the evaluation metric for the learned representation. However, the true cluster structure of the population is unknown. and the typical downstream task in representation learning is often prediction. I suggest that the authors design a downstream task, such as genomic prediction, to better evaluate the performance of the learned representation.**

Response: Our main goal in creating the embeddings was to be able to produce visualizations that represent the study population well. We intend to use the classification accuracy only as a tool to evaluate how the embedding retains the population structure. With that, the main goal was not the population classification task itself, but we think the way we are using population classification is a good way to evaluate the quality of the embeddings, assuming that the embeddings are created in an unsupervised/self-supervised manner without access to labels. Some populations might not actually be genetically homogenous, so a perfect score is unlikely, but if a method successfully cluster based on genetic similarity in such a way that individuals with identical labels end up close together, that indicates the detection of a relevant signal.

In the future, we want to explore using these methods for embeddings of higher dimensions for use in tasks including genomic prediction, but as it stands now, our methodology is tailored towards performing well on low-dimensional (visualizable) embeddings. Like the reviewer mentions, the literature on representation learning is mostly using higher dimensional embeddings (128 or 1024, for example) for use in prediction tasks, but considering the considerable use of dimensionality reduction purposes for data visualization and exploration in genetics, we believe that new developments within that realm are also of relevance. Therefore, our edits to the manuscript have focused on improving the experiments comparing our method to our approaches in the low-dimensional scenario.

1.2. **Regarding the data augmentation, I think masking SNP data to generate positive pairs is a good idea. However, I'm unclear on the rationale for flipping SNPs. Flipping a SNP can result in a**

completely different genetic meaning. It would be helpful if the authors could add a paragraph discussing this choice.

Response: We agree that the masking of SNPs is a more natural augmentation choice, since just masking some SNP will in some sense just be a lower quality representation of the same genotype, while flipping some markers actually changes the genotype. We are more interested in finding and representing haplotype blocks and flipping some of the markers will still retain short-range haplotypes. We have updated the manuscript both with a paragraph explaining the reasoning a bit more, and with an empirical ablation study over the two augmentation parameters p_{mask} and p_{flip} that shows that marker flip is (perhaps surprisingly) the stronger contributor to overall classification score as compared to the masking augmentation. Shown in Table 1, and discussed at [P. 5, L. 51 | P. 5 L. 69]

1.3. **When generating positive pairs, how is the proportion of masked alleles controlled?**

Response: The proportion of masked alleles is controlled by the hyperparameter p_{mask} . For each training batch, the proportion of alleles that will be masked is drawn from a uniform distribution between 0.001 and p_{mask_max} . The masking is done anew in each iteration, so the same samples will have a different number of markers masked each time it is seen by the model. This has been made clearer in the section *Augmentations as positives*. [P. 5, L. 73 | P. 5 L. 91]

1.4. **The novelty of this method seems to be the application of the centroid idea to the N-pair loss. I wonder if this approach could also be applied to other well-known contrastive loss functions, such as the NT-Xent loss.**

Response: Yes, the centroid version of the N-pair loss is to our knowledge novel to this work, we have also not seen the masking augmentation scheme used in this setting. The augmentation scheme and the loss used can be seen as two distinct modules in this framework. Both different choices of augmentations and loss functions can in theory be used interchangeably. Common losses in contrastive learning typically use larger output dimensions of say 128-256. In this work, we want to have a lower output dimension, so that we can visualize the embedding directly. We have tried different contemporary losses in this low-dimensional embedding model, but they did not perform well in our experiments. This is when the idea of the centroid-based version came up, which we saw performed better.

1.5. **In equation 3, where the centroid is computed, the negatives are multiplied twice. How is the weight of the negatives determined? Is this adjustable, and can it be treated as a hyperparameter?**

Response: The choice to weight the negative twice came from an assumption that in general, at least after some training, the positive and the anchor will most likely be mapped relatively close. Furthermore, since the positive and the anchor represent the same individual, weighting the negative by two makes the centroid represent the proper centroid between individuals. With this configuration in mind, it feels natural to have the centroid in the middle of the three points, illustrated in Figure R1. This also reveals the reason behind the subtraction of $\exp(-2)$ in Equation 6: If the positive and the anchor are mapped to the same position, we should have a loss of zero. The dot product with respect to the centroid of the positive and anchor, and anchor and negative, will be 1 and -1, respectively. If the anchor and positive are not close, this weighing will result in enhancing the repulsion from the negatives more, which we do not think is a bad property but might rather help in early optimization.

While this scaling could be treated as a hyperparameter that balances the attraction to positives versus the repulsion from negatives, we think that the current choice is a natural one. We have made adjustments to the manuscript to better describe this [P. 4, L. 29 | P. 4 L. 43].

2. **Reviewer 2:** *In this paper the authors develop a frame work for contrastive learning in population genetic data. Briefly, the authors utilize a data augmentation approach for calculating various contrastive loss functions in the context of a deep convolution architecture that takes as input genotype data that with a masking channel. This sort of contrastive learning is encoder only and thus yields an embedding space of the data, which the authors then describe with a 2D visualization and KNN classification metric. Comparisons are made to embeddings made with PCA and t-SNE. The results are interesting and the discussion thoughtful and well-pitched.*

2.1. **The authors evaluate the model on two datasets- one from human and one from dogs. While evaluation on empirical data is the ultimate goal, it would be helpful to also apply the contrastive learning method on simulated data. In particular multi-population simulations could be**

Figure R1: Position of the centroid based on different scaling of the negatives.

performed in the context of the coalescent where the 'ground truth' would be known more fully. A comparison of genealogical distance in the simulated ARG versus learned embedding space could be particularly interesting to compare, in addition to the metrics and visualizations the authors are already using. An easy way to accomplish this would be to use `stdpopsim`.

Response: We think that the results we show on empirical data are convincing enough to show both the performance and a more interesting case than simulated data. We have updated the metrics we use, and also compare the embedding distances with distances in genotype space which gives a more nuanced interpretation of the embedding qualities. The experiments done for the two empirical datasets we have used have been extended. Simulated datasets would be a welcome addition, but we still think that the two datasets we use have relatively different characteristics when it comes to population structure, and should illustrate how the methods handle these differences.

In our masking experiment, we also have a known ground truth for the masked individuals. We can therefore assess the level of changes induced in this scenario.

2.2. In addition to the comparisons being made here, it would good to compare the learned embeddings to those from earlier autoencoder methods like GCAE or popvae. How much better is this new method? How do training costs/times compare?

Response: We have now included popvae in the comparison of methods, we compare both in terms of embedding quality and training times. The performance is compared throughout the Results section, and timings are compared at [P. 15, L. 32 | P. 15 L. 85]

2.3. The data augmentation scheme requires some arbitrary choices. What are the effects on learning of the inclusion of flipping? What does performance look like without it? What effect does the hyperprior on the prob(missing data) have? Perhaps some simple supplemental tables can be adding showing comparisons of loss with different values of `p_flip` and `p_mask`?

Response: This is a good point, also raised by reviewer 1. We have made changes to the manuscript, refer to point 1.2 for some reasoning behind the augmentation, and where changes have been made.

2.4. The embedding has output dimension = 3, but for visualization the authors do a 2D projection. What is the effect of changing the output dimensionality, in particular to 2? What about increasing the embedding dimensionality? Do higher dimensional embeddings lead to greater KNN accuracy? What do 2D projections of higher dimensional outputs look like? **Response:** Figure S2 in the supplementary shows how the 3NN accuracy changes when going from the 3D sphere to the

~~1st Revision - Authors' Response to Reviewers: January 21, 2025~~

2.5. ~~2.5. Some typos using the map projection on the human origins data. In the experiments, we report the kNN accuracy on the sphere, to capture cases where populations are split by the projection, like the Tibetan Terrier clade in the dog dataset, seen in Figure 6.~~

With regards to higher dimensional embeddings, as described in response 1.1, we have limited our analysis to low dimensional representations. Our main goal is visualizing population structure, and not maximizing population classification performance. The centroid loss is designed specifically for this use case.

The 2D projection we apply is the Equal Earth map projection. This only works for projecting 3D spheres down to 2D, for projection from larger dimensional outputs other methods are needed.

- 2.5. **some typos: 1. -L19P4 - typo "but be" repeated, 2. -L30P6 - typo missing word, 3. -L49P6 - reference to McVean out of place**

Response: These typos have been corrected, thank you. 1: [P. 4, L. 66 | P. 5 L. 6], 2: [P. 7, L. 23 | P. 7 L. 42], 3: [P. 7, L. 44 | P. 7 L. 61]

- 2.6. **I'm a bit confused about the inclusion of the 'run_Cl.py' script and the MNIST image examples. This seems like a separate project no? It's not clear what's going on with that section of the readme**

Response: The intention with the examples on image data is to provide another example application using the centroid loss function. We have moved the image example and the parts in the readme relating to the image application to its own folder, to make it less confusing.

- 2.7. **I didn't have lunch running the code on my system I was getting an error of the form: ' len is not well defined for a symbolic Tensor (autoencoder_1/sep_sparse_oh_1/Shape:0). Please call 'x.shape' rather than 'len(x)' for shape information.' I was forced to use a slightly different tensorflow version for the singularity app (tf 2.18) so perhaps this could explain the error. The nvidia container the authors are using uses an older tf version, so it would be great if they could support newer tensorflow versions in their revision.**

Response: As you noted, this error was due to a version discrepancy. We have made the necessary adjustments to the code to fix the error you saw, and were able to run with Tensorflow 2.18. However, the code was developed for TensorFlow 2.15, which is still the recommended version for using the code. Using the supplied Apptainer image should work.

3. **Reviewer 3:** *In this manuscript the authors present a framework to use contrastive learning, a form of dimensionality reduction using neural networks, to visualize population structure. The manuscript provides in-depth detail on the method and tests it on two datasets (one for humans, one for dogs). In contrast to existing methods such as PCA, t-SNE, and UMAP—which use single nucleotide data—this approach uses convolutional layers which the authors state also may capture patterns of linkage. Their approach also uses a loss function defined specifically for SNP data. Computationally, the method is very slow, but can be sped up using parallelized GPUs.*

- 3.1. **This is a small comment but the authors alternate between “NN” and “neural network” throughout the manuscript. I suggest sticking with “neural network” as later in the manuscript they discuss KNN (k-nearest neighbours) and it can get confusing with “NN”.**

Response: This was a good observation. We have changed all the occurrences of "NN" to neural network throughout the manuscript for improved clarity.

- 3.2. **In the introduction they discuss the uses of t-SNE and UMAP in visualizing population structure. They also state that the usage of these methods has “sparked controversy and faced community criticism”. This is a valid statement, however it is not clear what the controversies and criticism they are referring to. This could possibly refer to the UMAP recently presented in the All of Us flagship paper (discussed in Kozlov, 2024, Nature); alternately, it could refer to some of the critiques specific to neighbour embeddings or to the methods specifically (e.g. that they do not have interpretable long-range distances). I believe this is important since it motivates the rest of the manuscript. If it is in response to how race, ethnicity, and genetics are presented in figures, I do not think that a new method adequately addresses the problem regardless of whether it has robust theoretical properties, particularly if it is motivated by population classification. If it is with respect to addressing specific methodological shortcomings of PCA, t-SNE, UMAP, etc., then the authors must be explicit about the shortcomings and how their method seeks to address them**

We thank the reviewer for the comment. First of all we argue that there are important conclusions to draw regarding population structure and that data with labels can certainly have a role. We were mainly referring to the All of Us controversy, where we found the combination of using socially charged labels and an embedding that promoted disconnected clusters was especially disconcerting. This part of the introduction has been rewritten, starting at [P. 1, L. 43 | P. 1 L. 47].

As we then get back to demonstrate in Figure 10, UMAP and t-SNE will both indeed have a tendency to completely lose some parts of the neighborhood structure, while at the same time (as indicated by the other metrics) performing very well in preserving it overall. Qualitatively, this can mean that an admixed individual is placed squarely in one cluster. Our proposed method performs far better than PCA on several of the other metrics, while still having very similar neighborhood-preserving power compared to PCA. We think that the property of a globally additive embedding can sometimes be worth sacrificing, if we preserve adjacency and similarity neighborhoods for all individuals that actually exist in a dataset. This interpretation of Figure 10 is addressed in the discussion, [P. 14, L. 18 | P. 14 L. 58]. Outside of the scope of this paper, we have also experimented with augmentation strategies actively introducing simulated admixed individuals to ensure that they are properly embedded. That work was too preliminary to be reported here.

- 3.3. **Throughout the manuscript they describe one of the shortcomings of t-SNE and UMAP as methods that are unable to project new data onto an existing embedding in a straightforward manner. UMAP has long had this ability in its main Python implementation (via the fit and transform methods); the scikit-learn version of t-SNE does not have this ability, but it is available in newer packages (e.g. openTSNE). If it is a computational or theoretical issue (e.g. it is overly heuristic), the authors should clarify this point.**

Response: This ability was overlooked. Now, all methods in the experiments produce the full embeddings by training on the same set of training individuals, and only after training transforming the validation samples.

Still, there is still a major difference in the way projection of new samples works that we think is important. t-SNE and UMAP are neighbor-based methods, and projection includes comparing the new (validation) samples to the existing ones (training samples). This comparison is done in both the embedding space and the input space. The implementations mentioned in this point for t-SNE and UMAP both save and use the raw input data in their respective instantiated objects, (`umap.raw_data`, in the case of UMAP). Changing the value of this saved raw data after training changes the resulting projection. In contrast, to embed new samples with a trained PCA embedding, we only need the weights of each input SNP to compute the linear combination which constitutes the embedding. Similarly, to embed new samples with a neural network model, we only need the model architecture and the weights.

This makes a big difference both in terms of data volumes, and privacy preservation. While privacy can not be guaranteed to be preserved for higher dimensional PCAs, nor for a trained neural network model, there is at least no need to distribute the training data which may be both sensitive and large.

We have made changes in the manuscript to better describe this issue, rather than saying it is not possible [P. 2, L. 62 | P. 2 L. 76].

- 3.4. **The authors compare their method to t-SNE and PCA throughout the manuscript; I believe it would also be appropriate to compare to UMAP. Using their method, they project data to a three-dimensional sphere which they then flatten. While t-SNE cannot do this, UMAP has the ability to specify the type of space to which data are projected (see documentation). It would be worthwhile to make this apples-to-apples comparison, particularly since I haven't seen others take this approach to 2D visualizations.**

Response:

We have now added UMAP in the comparison in the manuscript. We tried using the spherical embeddings for UMAP, but found that they performed worse than plain 2D UMAP (at least under default parameters). We have included the performance metrics for this experiment in the Supplementary in Table S1, and added a short paragraph in the manuscript [P. 9, L. 26 | P. 9 L. 58].

- 3.5. **Given that it is a neural network approach, I think it would be important to compare to existing methods that they cite (such as GCAE), if not as part of the analysis then at least citing its performance relative to GCAE.**

Response: We have now included experiments using the variational autoencoder model popvae to compare our method to another neural network model.

- 3.6. **For t-SNE, it would be helpful to specify the parameters used (particularly perplexity). This**

could go in the supplement if necessary.

Response: We have now included better information about the settings used for both t-SNE, UMAP, and test both methods with 2 settings for the number of neighbors considered.[P. 8, L. 89 | P. 9 L. 1]

- 3.7. **The description of the method is quite long and comprises about half of the manuscript's text (approximately 3.5 pages of methods text vs 3.5 pages of other text). Some of this could be shortened or moved to the supplement (e.g. the portions about Model Architecture related to spherical embeddings or hyperparameters for the Adam optimizer).**

Response:

We have made substantial revisions to the experiments and the evaluation of the embeddings, which we think balances the paper better. While the methods section might be a bit heavy, we think that it is important to keep it thorough to properly introduce the average Genetics reader to methods they might be unfamiliar with, to lessen the risk of misunderstandings.

- 3.8. **The Human Origins data is available online, but it would be nice to have a direct link to it along with the citation. The dog dataset is linked on their github repo.**

Response: We have added a link to the webpage of the human data [P. 6, L. 50 | P. 7 L. 4].

- 3.9. **The results about how to code missing data (-1 vs 9 vs imputing the most common variant) are minor but still interesting.**

Response: We appreciate the comment. After changing to training and then projecting with the OpenTSNE API, while the best way to do it is still imputing to the most common genotype, the results are not as visually striking anymore. We have thus decided to remove this section from the supplementary text.

- 3.10. **The contrast between methods in Figure 8 is also interesting and I am curious if there are any consequences to the (possibly divergent) conclusions you could draw from comparing these different clusterings. This seems like an important difference but it is not deeply explored.**

Response: We agree that the previous version of the t-SNE graph was qualitatively very dissimilar. However, when applying the more reasonable tuning now used for t-SNE and UMAP, as a result of the responses to the other comments, the difference is now far less striking. We get a similar amount of empty space between superpopulations in all methods used. For the veracity of those clusters, and especially the issue of whether genetically highly similar individuals would still end up with considerable space between them, we refer to the new Figure 10 and the accompanying discussion. The text has also been clarified to point out the qualitative behavior associated with the panel masking in the right-hand column for the various methods.

- 3.11. **For the data availability, please add the URLs for the data rather than referring readers to citations.**

Response: Now both the human and dog dataset are available by URL links [P. 6, L. 50 | P. 7 L. 4].

- 3.12. **1 – Line 45: As mentioned above, be specific about what the controversies and criticisms are.**

Response: Previously addressed in 3.2 [P. 1, L. 43 | P. 1 L. 47].

- 3.13. **2 – Final two paragraphs: Can this method actually preserve patterns of linkage? This seems to me like an important strength that would set it apart from SNP-based methods but is only mentioned in passing as a possible theoretical benefit. This type of analysis has been done before (see e.g. Battey et al, 2021, Fig. 9)**

Response: We mostly opted for using convolution layers based on the general argument of being able to capture linkage, as reported in Ausmees and Nettelblad [2022]. As we note in the manuscript, even more strongly in the edited version, we believe there to be considerable room for further optimization of model architecture, but still believe that convolution or alternatively attention-based approaches with position encoding, should be of relevance genotype data. We have toned down the direct claims of capturing linkage patterns in the manuscript, since it is not a core part of the analysis within this paper [P. 2, L. 112 | P. 3 L. 4].

The way Battey et al quantifies how well their method captures LD patterns consists of inspecting the output genotypes. They do this by computing the LD of the output (genotype reconstructions), and comparing it to the LD of the true genotypes. This type of analysis can not be done with our framework, since our method does not produce any output genotypes, but only an embedding. It is only an encoder, lacking the decoder part of autoencoder models. In other contrastive applications, auxiliary decoders have sometimes been trained

3.14. **2 – Line 105: “in a way an indirect way” is unclear**

Response: This has been shortened to be more concise, thank you. [P. 3, L. 48 | P. 3 L. 61].

3.15. **Figure 1: Some typos in the last sentence. It should read “In both models, the embedding coordinates are used as the population visualization.”**

Response: The typos have been corrected, thank you. (Figure 1 Caption)

3.16. **3 – Line 5: Just to be clear, a “batch” is a group of individuals rather than a collection of SNPs, correct?**

Response: Yes, this is correct. This has been clarified in the text [P. 3, L. 69 | P. 3 L. 82].

3.17. **3 – Line 10: I would split this sentence as it is unclear. I would also recommend defining “anchor”, “positive”, and “negative” for readers unfamiliar with machine learning terminology.**

Response: We have added better introduction to the anchors, negatives, and positives, and made the section more clear. [P. 3, L. 75 | P. 3 L. 88] and [P. 3, L. 81 | P. 3 L. 97].

3.18. **4 – Line 75: I’m unclear on how the data are encoded. Does pflip determine the type of flip (e.g. 1 to 0 vs 1 to 2)? Or rather just that it will be flipped? Also how does the system determine the direction of a flip for the value 1? E.g. does it go 1 to 2 or 1 to 0?**

Response:

The parameter p_{flip} determines the proportion of markers to be flipped in the input genotype. The flip is always done in one allele, so that 0 always flips to 1, 2 always to 1, and 1 flips to either 0 or 2, with equal probability. This is done to keep the flipping “mild”. We have changes to the manuscript to make the description more clear [P. 5, L. 47 | P. 5 L. 65].

3.19. **4 – Final two paragraphs: The order of the text suggests that you first mask and flip, while the figure illustrates that you flip and then mask. I suspect the latter is the case; if so, I would re-arrange the text to be clearer. I would also be explicit about what the onehot encoding does. It is mentioned in the text and left to the reader to infer based on Figure 3.**

Response: Yes, in the code we first flip, then mask. This is now better reflected in the manuscript where we explicitly state this order. [P. 5, L. 59 | P. 5 L. 77]. The one-hot encoding is now more explicit [P. 5, L. 62 | P. 5 L. 80].

3.20. **Figure 3 – This figure introduces the term “sparsify”, which I believe refers to the masking. As far as I can tell this is the only usage of the word “sparsify”—it would be clearer to write “masking” (or to define “sparsify” in the text somewhere). The one-hot encoding is not obvious at a glance, though I’m inferring that 1,0,0,0 represents unchanged data, 0,1,0,0 represents a SNP flipped to “1”, 0,0,1,0 is a SNP flipped to “2”, and 0,0,0,1 is a SNP that has been masked. This explanation would be useful in the text or caption.**

Response: The term sparsify is an artifact from the code, and has been changed to mask in Figure 3 for coherence with the text. As submitted, the one-hot encoding was open for misunderstanding. We have now added a more explicit description of how this is done [P. 5, L. 62 | P. 5 L. 80].

3.21. **Figure 4 – The flow in this figure needs to be tidied. Text next to the arrows indicates both data (“Input data”) and processes (SiLU). It should finish with “Output data” or something similar, as it currently has an arrow with “L2 normalization” next to it and points to the caption text.**

Response: The illustration of the model architecture has been made more clear and readable in Figure 4.

~~3.22.3. Line 10: I'm confused about how you choose samples randomly based on the inverse embeddings distance given that the embedding is generated at the end of the process. Is this done with an initial embedding before optimization, or its distance in the highdimensional space?~~

Response: In short, training a neural network consists of the following steps

- Generate a batch of input data x , consisting of a set of training samples
- Compute a forward pass of the model, generating outputs z
- Compute the loss function given the output $z \leftarrow$ *here the negatives are chosen based on z*
- Compute the gradient of the loss function with respect to the network weights
- Update the weights of the network using the gradients, and the chosen optimizer.
- Repeat the above steps until convergence.

So, yes, the embedding is produced at the end of the model, but during training, computing the embedding is the second step. The choice of negatives based on the inverse distance is done when computing the loss function, before computing the gradient and updating the weights. So the choice of negatives is done on the embedding coordinates in the current training batch, which are available during training.

We have updated the text for clarity [P. 5, L. 88 | P. 6 L. 8].

3.23. **5 – Model architecture: I believe UMAP would be a good comparison to this approach rather than t-SNE. It is fairly straightforward to project to a 3D unit sphere and I think this would be a very interesting result.**

Response: We have in our revised manuscript added results for UMAP as well as popvae. The UMAP 3D projection an interesting comparison to do. In our experiments, the spherical UMAP performed worse than the normal 2D UMAP in the label-based metrics we have used. The metrics for the spherical UMAP is compared to 2D UMAP in the supplementary text, Table S1, . A small paragraph at [P. 9, L. 26 | P. 9 L. 58].

3.24. **Figure 5: Typo in “This to mimic samples”** This typo has been corrected, thank you [Figure 5 caption].

3.25. **6 – Performance metrics: Stray citation to McVean 2009 at the start of the section.** The reference has been corrected, thank you [P. 7, L. 44 | P. 7 L. 61]

3.26. **6 – 57: Sentence is confusing. I understand the point but it could be simplified to be clearer.** This sentence has been clarified, thank you. [P. 7, L. 52 | P. 7 L. 76]

3.27. **6 – 60: This sentence also needs simplification as the word “sample” is playing three different roles.** The sentence has been reformulated for clarity. [P. 7, L. 56 | P. 7 L. 81]

3.28. **Figure 6: It is very difficult to make out any details in the figure because there are so many combinations of shapes and colours. Consider collapsing some of the categories.**

Response: We agree that with the combinations of color and shapes might make it harder to get an overview of the population structure. However, we think that it is informative to show the population label using the more fine-grained breed level labels, which would be lost if they were only plotted by the clade label. To make it easier to follow the points made in the text when mentioning different clades, we have highlighted the clusters in Figure 6 that are talked about in the discussion.

3.29. **8 – 31: Is this a consequence of your contrastive learning approach? Or is this the result of choosing to embed your data in a 3D sphere and then flattening that into 2D?**

Response: The embedding being on a continuum, and not bound to a center of gravity could be said to be a consequence of both our loss function and the spherical embedding. Embedding samples on the sphere ensures that samples will have neighbors in all directions, and no sample will lie "on the edge", or outskirts of the embedding space. The definition of our loss function makes comparisons of embedding coordinates in a local reference coordinate system, with no relation to some arbitrarily defined origin. An embedding with a strongly emphasized origin from which samples radiate might lead readers to believe a population close to the origin to be some sort of "reference", or founder population, where such a inference can not be made, as this might rather be dictated by the sample size of different populations. We have added a paragraph discussing this [P. 13, L. 2 | P. 14 L. 4].

~~1st Revision: Authors' Response to Reviewers, January 21, 2025.~~
3.30. **8 – 37: It is not clear where I am supposed to find the points in the t-SNE embedding. If you'd like to highlight specific points, I recommend adding a new panel where the other points are grey or more faint and the points you wish to discuss are clearly highlighted. I am also not sure I agree with the interpretation... when a reader is presented with a figure, won't some cluster always be outlying?**

Response: We have highlighted the clades discussed in the text in the plots to make it easier to follow along with the reasoning in the discussion section [Figure 6].

Given that a reader knows that our embeddings are spherical, and wrap around the x-axis, we would argue that it is harder to identify some populations as outliers. By removing the origin that the 2D embeddings have, we can only really say something about a population with regards to how it is placed in relation to other populations, rather than if it has "extreme" coordinates. While it might be more unintuitive to view a 2D projection of the sphere, knowing that the embedding is spherical and has this wrapping property lessens the notion of a cluster being outlying based on its coordinate. We discuss this in [P. 10, L. 103 | P. 14 L. 4] as well.

3.31. **8 – 48: Why not compare in 3D? Visualization and classification are separate tasks and I would imagine that given the capability we would do classification in 3 dimensions rather than 2.**

Response: The main application we target here is population structure visualization. It is true that the classification should be better in 3D than in 2D, but here the main goal for the classification score to use it as a way of quantifying the visualization performance.

We have tried the other suggested way of comparing with spherical UMAP in 3D, but saw decreased performance compared to 2D. As described in the response to point 1.1, the main for the population classification is not as the target application itself, but rather to use it as a tool to evaluate the embeddings, since the label information says something about the population structure.

3.32. **8 – 86: "entire dataset" repeated. Response:** This line was removed in the revised version [P. 15, L. 51 | P. 15 L. 66].

3.33. **Figure 8: Do the different shades or shapes represent anything?**

Response: Yes, samples with the same shade represent samples from the same superpopulations, and samples with the same shade and shape are from the same subpopulation. The text has been updated to better describe this [Figure 9 caption and legend].

3.34. **11 – 6: I believe you mean "transparent", not "translucent" Response:** Thank you, this has been corrected. [P. 15, L. 56 | P. 15 L. 115]

4. Associate editor comments:

4.1. **I agree with the reviewers that the performance evaluation needs more expansion to evaluate if the methodology is sufficiently an advance over previous work to be relevant to practicing geneticists. For the classification task presented, 2D-PCA will clearly do poorly in classification settings with more than 2 populations, and tSNE is known to have disadvantages relative to popvae and UMAP - so I agree with the others that the experiment is not yet complete/satisfying. For evaluating the performance more thoroughly - besides comparing to UMAP and popvae, you could take inspiration from the Chari and Pachter (PLoS Comp Bio 2023) and carry out experiments to show how the method performs at preserving: 1) Local structure; 2) Global structure; 3) Distance in the multi-dimensional space.**

Response: Most of our edits to the manuscript have focused on providing more relevant comparisons, both in terms of the exact methods we compare against and the metrics reported. We appreciate the suggestion to be inspired by Chari and Pachter, which we have implemented, focusing on modified versions of their local metric. Their suggested global metric relies heavily on the labels, and while we do have labels in our data, the local structure metric had the distinct advantage of showing behavior independent of the labels. We therefore added this metric, since it complemented the kNN label conclusions already reported. By scaling the number of neighbors considered in this "local" metric, it does in fact scale to showcase global behavior.

We have opted not to directly consider the exact recreation of distances or their ranking, while this is present to some extent in the local metric. Our reasoning for this is also mentioned in the paper. The exact marker set included will directly influence distances. Depending on the normalization scheme used, by definition no method will be superior to PCA in recreating plain Euclidean distances. On the other hand, just doing

1st Revision: Authors' Response to Reviewers: January 21, 2025

~~different but interesting, of the same dataset, we produced two different induced distance metrics, while describing the very same population. We believe attempts to directly reconstruct distances expressed in this way are misdirected. However, we find it crucial that genetically similar individuals should always end up together, and that reasonably similar individuals should end up reasonably close. We have therefore introduced Figure 10, where we investigate to what extent the different methods reconstruct the relative ranking of the nth neighbor with n ranging from 1 to 100. This figure captures the fact that t-SNE and UMAP are both prone to break some adjacency relationships while maintaining others. Our contrastive method performs very close to PCA, with popvae lagging slightly behind.~~

The changes relating to this point are many, and hard to pin down to a line number. They are mostly found in the "Performance metrics" section from "label-based metrics", down to the results on the dog dataset. This point has also influenced how the results are presented, and added Figures 7 and 10.

- 4.2. **There is a sense in the reviews that the paper is imbalanced with relatively few results for a long methods section. I did not mind the longer methods as it was explanatory and did a good job at that. Hopefully adding more results will balance the manuscript. That said, I could also see strategically taking the more didactic components of the methods section and moving them into a supp info section titled akin to "Introduction to Contrastive Learning" such that your main text can tell on the specific innovations / choices made in your team's implementation of a contrastive learning approach for dissecting population structure.**

Response: We appreciate the comment. We also note that some comments from the reviewers might be interpreted as misunderstandings about the nature of the contrastive approach. We believe that the risk for such misunderstandings would be even greater, especially for a less informed readership, if we would cut down the current explanation. We have made attempts to improve the methods presentation somewhat. The significant increase in quantitative results, including the inclusion of UMAP as well as popvae, and the additional metrics, in our opinion also restores some balance to the current manuscript.

References

Kristiina Ausmees and Carl Nettelblad. A deep learning framework for characterization of genotype data. *G3 Genes—Genomes—Genetics*, 12(3):jkac020, 01 2022. ISSN 2160-1836. doi: 10.1093/g3journal/jkac020. URL <https://doi.org/10.1093/g3journal/jkac020>.

February 24, 2025

RE: GENETICS-2025-307812

Dear Dr. Thor:

I am pleased to accept your manuscript titled "Dimensionality Reduction of Genetic Data using Contrastive Learning" for publication in GENETICS, pending minor revision.

You'll see there are some very minor points raised with a remaining request to see the method run on some basic simulations. You can read the comments below.

Once ready, please submit your revision along with a brief description in response to the reviewers' concerns and suggestions (which can be viewed at the bottom of this email).

Please ensure that you have included a Data Availability Statement at the end of the Materials and Methods section. Details available at <https://academic.oup.com/genetics/content/prep-manuscript>. The DAS should include the accession numbers or DOIs of any data you have placed in public repositories, describe supplemental material, include applicable IRB numbers, and may include specifications for how to properly acknowledge or cite the data.

When revising the ms., please make an effort to shorten it, because that almost always improves a manuscript. We urge authors to heed the advice of Strunk and White: "omit needless words"¹. Follow this link to submit the revised manuscript: Link Not Available

Thank you for submitting this story to Genetics.

Sincerely,

John Novembre
Associate Editor
GENETICS

Approved by:
Hongyu Zhao
Senior Editor
GENETICS

Reviewer comments:

Reviewer #1 :

Thank you for addressing my comments. The explanation on the SNP flip augmentation is really helpful. At this stage, my additional comments are only on minor grammar points.

1. In clean version page 1 line 44-45, "While they can group samples with similar features they, unlike PCA, do not preserve an interpretable distance between samples in the embedding." should be "While they can group samples with similar features; they, unlike PCA, do not preserve an interpretable distance between samples in the embedding."

2. In clean version page 13 line 19-20, "For example, in Figure 6 both PCA and popvae embeds subpopulations from ..." should be "For example, in Figure 6 both PCA and popvae embed subpopulations from ..."

Reviewer #2 :

I'm generally happy with the revisions the authors have made and think that it leads to a strengthened paper. Having said that I think the decision to not include simulated data is a mistake here, but I leave that to the discretion of the editor.

Reviewer #3 :

The manuscript has been greatly improved. A few minor comments:

- Abstract line 3: I think the correct grammar would be "Much of the advance" or "Many of the advances"
- Page 1 line 53 to Page 2 line 2: I don't understand what this sentence is meant to say. Regardless we don't know how participants would socially identify their heritage (this is a limitation inherent from providing a few census categories) so it may be worthwhile to cut the sentence.
- Page 2 line 26: Isn't this a consequence of the data as well? I imagine an admixed family (e.g. children whose mother and father come from different continents) could never satisfy this criterion. Clustering also suffers from this problem in general since it is not possible for a cluster to both contain all similar data points and not contain all dissimilar points. I don't think this needs to be edited, just a thought.
- Page 2 line 37: Is "origin" specifically referring to the race/ethnicity variables in All of Us? Or is this a more general point -- I'm trying to make sure "origin" does not get conflated with race/ethnicity variables. Perhaps "demographic data" would work better than "self-reported origin".

Other notes:

- Website citations (e.g. citations of tweets) should be archived via services like archive.org or archive.today to avoid link rot
- There is a parametric version of UMAP that uses neural nets and can be used to transform new data (https://umap-learn.readthedocs.io/en/latest/transform_landmarked_pumap.html; <https://direct.mit.edu/neco/article/33/11/2881/107068/Parametric-UMAP-Embeddings-for-Representation-and>). A comparison would probably be of interest to the authors but out of scope for this manuscript

Associate Editor comments:

Reviewer 2 mentions simulations as a lingering reservation. I agree that the empirical datasets you've used and the metrics you've provided demonstrate advantages of your method. That said - I do think it's prudent to do a few basic simulations to check against any surprises or unexpected behaviors (I have a paper from 2008 on artifacts that arise in PCA with spatial structure that emerged from running PCA on simple simulations that should have been run years earlier by others before my time [Novembre & Stephens 2008 Nature Genetics]). To keep it simple my two go-to's for a quick sanity check using simulations would be something like: A) two population split for 0.5 N generations followed by 80/20 admixture of individuals 5 generations ago; Samples come from the two source populations and the admixed individuals with a sample size in the first source population that is twice that of the admixed sample and second source. B) A linear stepping-stone model with 10 demes and $Nm = 1$. Samples come from all populations equally sampled and in a second run, an uneven, clustered sampling from demes 1-4 and demes 7 and 8. If you wanted to add a supplemental section on these simulations, we'd welcome it and I would give it a quick turnaround. I'm also comfortable if you share the results in a response to us but do not update the manuscript. The goal is to confirm for all our interests that the method behaves as expected with no surprises or new cautionary advice that is needed for users. We mainly want to avoid follow-up papers by others that do the simulations and write critiques based on surprising or unexpected behavior from these scenarios. The request is not for an extensive simulation study - but we believe it is a good exercise.

As an alternative to my suggested simulations - McVean's 2009 PLOS Genetics paper which also focused on PCA also has some nice test scenarios for dimensionality reduction methods. Also if you don't know it, the python implementation of msprime and the stdpopsim catalog are helpful for configuring simulations quickly.

Finally - as a refining point - while I understand R2's suggestion - I don't think you need to investigate how well the methods recover features of the ARG as mentioned in R2's original note. As I mentioned above - I think you've shown advantages - the simulations will just help clarify if the methods, for instance, behave like PCA in terms of its sensitivity to uneven sample sizes and whether we should add a cautionary note to users about that.

Review response: Dimensionality Reduction of Genetic Data using Contrastive Learning

Filip Thor, Carl Nettelblad

Overall summary

We would like to thank the reviewers for their feedback, and the specific comments on the cases of simulated data from the editor. The changes from the previous version consist of minor edits in the text as suggested by the reviewers, and the additional experiments in the Supplementary. We have also tried to make the introduction and Methods section more concise.

We reference the line number in the clean version when noting where the changes have been made, and the edits are indicated in the version showing the changes.

Review responses

1. Reviewer 1: *Thank you for addressing my comments. The explanation on the SNP flip augmentation is really helpful. At this stage, my additional comments are only on minor grammar points.*

1.1. **In clean version page 1 line 44-45, "While they can group samples with similar features they, unlike PCA, do not preserve an interpretable distance between samples in the embedding." should be "While they can group samples with similar features; they, unlike PCA, do not preserve an interpretable distance between samples in the embedding."**

1.2. **In clean version page 13 line 19-20, "For example, in Figure 6 both PCA and popvae embeds subpopulations from ..." should be "For example, in Figure 6 both PCA and popvae embed subpopulations from ..."**

Response: Thank you, we have made alterations to the manuscript, and included both of these improvements. [Clean version page 1, line 44-45, and page 10 line 105]

2. Reviewer 2: *I'm generally happy with the revisions the authors have made and think that it leads to a strengthened paper. Having said that I think the decision to not include simulated data is a mistake here, but I leave that to the discretion of the editor.*

Response: It is a fair request to ask for cases of simulated data. We have included embeddings using simulated data in the appendix using the suggestions of specific cases from the editor, including a simple admixture case, and variations of a linear stepping-stone model. We believe that this shows that our method does not have any major unwanted artifacts in cases where we know the population structure, and supports the findings of some of the properties of other methods that we saw in the empirical data. We believe this result strengthens the case for our method. Thank you for being persistent in asking for these experiments.

3. Reviewer 3: *The manuscript has been greatly improved. A few minor comments:*

3.1. **Abstract line 3: I think the correct grammar would be "Much of the advance" or "Many of the advances"**

Response: Thank you, we chose to change this to "Many of the advances...". [Abstract line 3]

3.2. **Page 1 line 53 to Page 2 line 2: I don't understand what this sentence is meant to say. Regardless we don't know how participants would socially identify their heritage (this is a limitation inherent from providing a few census categories) so it may be worthwhile to cut the sentence.**

Response: We agree that this sentence might not be clear, and have decided to cut it. [Page 1, line 53]

~~3.3. Page 0 line 20 - Is it this a consequence of the data or not? Imagine an admixed family (e.g. children whose mother and father come from different continents) could never satisfy this criterion. Clustering also suffers from this problem in general since it is not possible for a cluster to both contain all similar data points and not contain all dissimilar points. I don't think this needs to be edited, just a thought.~~

Response: Perhaps this could have been worded clearer. While an admixed family like you describe would have a high haplotype similarity, they would likely not have a high genotype similarity, which is what we refer to here. But this is a good point, and worth considering, thank you.

3.4. **Page 2 line 37: Is "origin" specifically referring to the race/ethnicity variables in All of Us? Or is this a more general point – I'm trying to make sure "origin" does not get conflated with race/ethnicity variables. Perhaps "demographic data" would work better than "self-reported origin".**

Response: We agree with this point, and have opted to replace "self-reported origin" with "demographic data". [Page 2, line 33]

3.5. **Website citations (e.g. citations of tweets) should be archived via services like archive.org or archive.today to avoid link rot**

Response: This is a very good point, we have archived the tweet and referenced this instead.

3.6. **There is a parametric version of UMAP that uses neural nets and can be used to transform new data https://umap-learn.readthedocs.io/en/latest/transform_landmarked_pumap.html <https://direct.mit.edu/neco/article/33/11/2881/107068/Parametric-UMAP-Embeddings-for-Representation-and>). A comparison would probably be of interest to the authors but out of scope for this manuscript**

Response: A comparison would indeed be interesting. However, since it attempts to solve the same optimization criterion as the regular UMAP implementation, we would assume that the resulting embeddings carry the same properties. By this, we mean good local performance on a local scale as it showed in all our experiments, but would most likely still perform bad in the neighbor rank score shown in Figure 10. As hinted in the point though, it might be better at transforming new data, but we argue that this is not the main downside, when applied to genotype data, and we leave this comparison for a further study.

March 17, 2025

RE: GENETICS-2025-307812R1

Dr. Filip Thor
Uppsala Universitet
Department of Information Technology
Lägerhyddsvägen 1
Uppsala, N/A
Sweden

Dear Dr. Thor:

Congratulations! We are delighted to inform you that your manuscript titled "Dimensionality Reduction of Genetic Data using Contrastive Learning" is acceptable for publication in GENETICS. Many thanks for submitting your research to the journal.

To Proceed to Production:

1. Format your article according to GENETICS style, as discussed at <https://academic.oup.com/genetics/pages/general-instructions>, and upload your final files at <https://genetics.msubmit.net>.
2. Your manuscript will be published as-is (unedited-as submitted, reviewed, and accepted) at the GENETICS website as an Advanced Access article and deposited into PubMed shortly after receipt of source files and the completed license to publish. Please notify sourcefiles@thegsajournals.org if you do not wish to publish your article via Advanced Access.
3. We invite you to submit an original color figure related to your paper for consideration as cover art. Please email your submission to the editorial office or upload it with your final files. You can submit a small-sized image for evaluation, and if selected, the final image must be a TIFF file 2513px wide by 3263px high (8.375 by 10.875 inches; resolution of 600ppi). Please avoid graphs and small type.

If you have any questions or encounter any problems while uploading your accepted manuscript files, please email the editorial office at sourcefiles@thegsajournals.org.

Sincerely,

John Novembre
Associate Editor
GENETICS

Approved by:
Hongyu Zhao
Senior Editor
GENETICS

note: Please add jnls.author.support@oup.com and genetics.oup@kwglobal.com (or the domains @oup.com and @kwglobal.com) to your email program's "safe senders" list. You will be contacted by both at various points during the production process.